# Modelling the Proton-Conductive Membrane in Practical Polymer Electrolyte Membrane Fuel Cell (PEMFC) Simulation: A Review

**DOI:** 10.3390/membranes10110310

**Published:** 2020-10-28

**Authors:** Edmund J. F. Dickinson, Graham Smith

**Affiliations:** National Physical Laboratory, Hampton Road, Teddington TW11 0LW, UK; graham.smith@npl.co.uk

**Keywords:** PEM, PEFC, PEMFC, ionomer, polymer electrolyte membrane, polymer electrolyte membrane fuel cell, proton exchange membrane, proton exchange membrane fuel cell

## Abstract

Theoretical models used to describe the proton-conductive membrane in polymer electrolyte membrane fuel cells (PEMFCs) are reviewed, within the specific context of practical, physicochemical simulations of PEMFC device-scale performance and macroscopically observable behaviour. Reported models and their parameterisation (especially for Nafion 1100 materials) are compiled into a single source with consistent notation. Detailed attention is given to the Springer–Zawodzinski–Gottesfeld, Weber–Newman, and “binary friction model” methods of coupling proton transport with water uptake and diffusive water transport; alongside, data are compiled for the corresponding parameterisation of proton conductivity, water sorption isotherm, water diffusion coefficient, and electroosmotic drag coefficient. Subsequent sections address the formulation and parameterisation of models incorporating interfacial transport resistances, hydraulic transport of water, swelling and mechanical properties, transient and non-isothermal phenomena, and transport of dilute gases and other contaminants. Lastly, a section is dedicated to the formulation of models predicting the rate of membrane degradation and its influence on PEMFC behaviour.

## 1. Introduction

In this review article, we summarise and evaluate the diversity of methods applied in the literature to describe theoretically the transport phenomena within a proton-conductive polymer electrolyte membrane (PEM, hereafter generally abbreviated to “membrane”), as applied in practical simulation methods for low-temperature polymer electrolyte membrane fuel cell (PEMFC) applications. Within the context of such applications, we identify the specific equations most often used, and sources of empirical experimental data for quantifying parameters for specific materials, especially membranes based on perfluorosulfonic acid (PFSA) ionomers, such as Nafion™.

We specifically place our focus on the bulk proton-conductive membrane which acts as a barrier to gas crossover in PEMFC devices. It is not our purpose to attempt a comprehensive review of the general literature on proton-conductive polymer electrolyte membranes, for which the reader is directed to the excellent and exhaustive 2017 review article by Kusoglu and Weber [1] as well as prior works correlating structural and chemical properties to performance characteristics [2,3,4]. Neither do we attempt to consider theories around the morphology and role of ionomer material in the context of the composite structure of catalyst layers, which remains an important open topic of interest, and is discussed elsewhere [5,6,7,8]. We will also avoid discussion of atomic-scale theories of the physicochemical structure of materials and instead focus attention on macroscopically observable transport behaviour of the membrane. We will avoid considerations specific to cold start (freezing) conditions, and also exclude the general field of high-temperature PEM devices—our range of consideration here spans conventional low-temperature operating conditions (broadly, 60 °C ≤ *T* ≤ 90 °C).

In speaking of “practical simulation methods”, we focus our interest upon models predicting the performance and overall electrochemical behaviour of a PEMFC, as well as the fundamental theories that most directly inform the continuum description of the membrane in full cell models. Electrochemical PEMFC models extend from empirical, lumped models to 3D models specifically resolving the dimensions of the various PEMFC components: bipolar plate (“land”) design, gas channels, gas diffusion layers (GDL), microporous layers (MPL), catalyst layers (CL), and membrane. The role of the membrane model is to correlate quantitatively the observed overpotentials due to membrane losses, and the membrane’s role in the cell water balance, to more fundamental transport laws in the membrane, which depend in turn on the environmental conditions. Such spatially resolved electrochemical simulations are valuable tools for PEMFC stack and component designers as they allow rational design of optimised components and configurations. Equally, they lend insight to researchers investigating the local conditions experienced by different materials and components within operating devices. Higher-dimensional models and those offering greater fidelity of description of the fundamental physical behaviour are expected to be more accurately predictive, providing greater descriptive granularity with respect to changes in operating conditions as well as with respect to exact locations within a cell or stack.

With these goals and restrictions in mind, this paper constitutes a focused review with the intention of acting as a useful digest for the present state-of-the-art in membrane modelling, from the perspective of practicing PEMFC simulation scientists. A number of prior reviews have usefully summarised historical progress and trends, and we recommend these as necessary reading for researchers active in PEMFC theory [7,9,10,11,12,13]; here, we emphasise a synoptic discussion and a survey of the most recent developments with specific focus on the membrane as a feature of the fuel cell.

The primary theoretical literature poses challenges due to inconsistency in notation between different authors, and a lack of traceability of parameterisation. We also recognise that validation of a complex model by means of a polarisation curve alone is often inadequate without further corroborating diagnostics. The detailed discussion of model validation for PEMFCs, as an exercise in general, again exceeds the scope of this review; by presenting theoretical descriptions together in a single source, however, we aim to set the stage for facilitated inter-comparison of models and accelerated implementation of new models for comparison with experimental data. In the Perspective section below, we present a sampled review of recent PEMFC simulations, in which it is demonstrated that even contemporary theoretical works depend heavily on theories and parameterisation established in the early 1990s on then-current membrane materials. For this reason also, we have taken this opportunity to present a collective review of both legacy and recent theoretical developments, as opposed to a selective review considering only the most recent work.

An account of simulation methods requires a summary of the essential transport phenomena to be described by the membrane model (Section 2). We then introduce the most basic level of practical description of the membrane, in the form of lumped and charge transport-only models (Section 3), before proceeding to consider the thermodynamics of membrane hydration through water sorption (Section 4) and the corresponding formulation of models combining charge (proton) transport with water transport (Section 5), optionally including phenomena specific to the interface between the membrane and its environment (Section 6). Thereafter, we describe specialised extensions to the core membrane models: hydraulic transport and membrane mechanics (Section 7); transient phenomena (Section 8); non-isothermal phenomena (Section 9); gas crossover and transport of contaminants (Section 10); and membrane degradation (Section 11). We then provide an outlook on continuing needs for theoretical work (Section 12).

Text abbreviations and symbolic notation used in equations throughout are summarised in the Table A1 and Table A2 in Appendix A.

## 2. Proton-Exchange Membrane: Role and Essential Transport Phenomena

### 2.1. Role of the Membrane

The purpose of the membrane in a PEMFC is to act as a barrier to gas transport, thereby preventing direct mixing of H_2_ and O_2_, and preventing electron conduction between the anode and cathode electrodes while acting as an ionic conductor via mobility of protons (H^+^). The main active component of PEMFC membranes is the proton-conductive polymer (ionomer) phase. Most commonly, the proton-conductive phase in PEMFCs is a PFSA. These polymers have a perfluorinated hydrophobic backbone connected to hydrophilic sulfonic acid groups that act as strong acids with very labile dissociation of protons. In the presence of water, mobile proton-carrying species (such as hydronium, H_3_O^+^) form, and the ionic conductivity of the membrane is increased significantly. Membrane hydration is essential for a practically useful proton conductivity to be obtained, so it is common for models to account for the variability of membrane properties with water content, and to describe water transport concurrently with proton transport.

The most well-known example of a PFSA is Nafion, originally developed in the 1960s by DuPont, and now a brand owned by Chemours (Wilmington, DE, USA) [14,15]. Other PFSA materials have been widely used in recent years, with the primary differences being the length and chemistry of the chain linking the backbone and acid groups [16]; these include Aquivion [17,18] (originally developed by Dow as Hyflon and now Solvay) and 3M materials [19,20]. Historically, membranes were made solely of thick extruded sheets of pure proton-conductive polymer; more recently, however, composite membranes with features such as non-conducting polymer reinforcements (e.g., GORE-SELECT materials [21,22]) and radical scavengers [23] (e.g., Nafion XL [24]) have become more common. Figure 1 summarises some of the key developments in membrane technology. While contemporary state-of-the-art commercial membranes bear little resemblance to their 1960s progenitors, either in form or performance, the underlying chemistry and physics remain largely the same.

In the PEMFC modelling literature, the significant majority of works address membrane materials in the Nafion family, but other PFSA-based materials, including reinforced membranes, can be treated through similar theoretical approaches, provided experimental data are available. We highlight that it is likely to be insufficient to, for instance, use historical data from experiments on Nafion 115 membranes in models of very thin, reinforced membranes using a different PFSA. Besides one exception in the recent Chinese-language literature [25], we encountered in the literature no instances of models explicitly accounting for the altered properties of thin composite materials used in state-of-the-art devices.

Fundamental research has also considered alternative membrane chemistries: for example, those made with non-fluorinated hydrocarbons [15], with multiple acidic head groups [26] or by the incorporation of new monomers into the backbone [27]. Again, it is likely that such materials can be treated through similar approaches to those developed for Nafion and discussed in this text, provided sufficient experimental data are available. The hydroxide-conducting membranes used in anion exchange membrane fuel cells feature significantly different transport mechanisms [28] and so any adaptation of the models described here to these materials must be made with great caution.

Similar PFSA-based materials to those used in PEMFCs are also used for the membrane in polymer electrolyte membrane water electrolyser (PEMWE) applications, and in alcohol-fuelled proton-exchange fuel cells (direct methanol and direct ethanol fuel cells, DMFC/DEFC). In these devices, one or both faces of the PEM is in contact with liquid water, altering the environmental equilibration of the membrane material compared to the PEMFC case, where both faces of the membrane meet a gas phase (notwithstanding condensation of liquid water in the CLs), and either face of the membrane may be partially humidified depending on operating conditions. We shall draw attention to the applications of the theories reported herein to PEMWE and DMFC/DEFC simulation, selectively and as appropriate.

### 2.2. Membrane Types and Fundamental Material Properties

Essential properties of the dry PEM material are its density *ρ*_dry_ and equivalent weight *M*_EW_, where the equivalent weight is the mass of polymer per 1 mol of sulfonic acid groups. It is common to report the available ion-exchange capacity (IEC) measured by titration (often as milliequivalents/g of the acidic functional group), which in an ideal condition is the inverse of the equivalent weight (IEC = 1/*M*_EW_) [29]. IEC can also be measured under specified hydration conditions, in which case it will generally differ from the maximum available IEC.

Nafion 1100 is a standard material with *M*_EW_ ≈ 1.100 kg mol^−1^ and *ρ*_dry_ ≈ 2050 kg m^−3^ [1,30]. Both the Nafion 11*x* and Nafion 21*x* series have *M*_EW_ close to this value, with *x* denoting the thickness of the manufactured membrane in thousands of an inch (10^−3^ in, “mil”). We note one recent model [31] giving the dry density as 1970 kg m^−3^, which is equivalent to the basis weight at 5% water content (50% relative humidity, *T* = 23 °C) given on the Nafion 115 data sheet [32]. In this context it is important to recognise that the basis weight and dry density are not equivalent concepts—there will exist a discrepancy depending on the degree of swelling with water uptake (see also Section 7.2, “Membrane Expansion and Mechanical Constraint”, below). Of course, the ideally dry condition is not encountered in the PEMFC context, and so densities of the hydrated membrane have more practical relevance. The role of water sorption on density is discussed further below (Section 4, “Sorption of Water”).

The concentration of sulfonic acid groups *c*_f_ in the dry membrane (or, as an inverse, the molar volume of the dry membrane ${\bar{V}}_{p}$) is defined as:(1)$$c_{f}={\bar{V}}_{p}^{-1}\equiv \frac{\rho_{\mathrm{dry}}}{M_{\mathrm{EW}}}$$

From the above standard values, *c*_f_ ≈ 1850 mol m^−3^, ${\bar{V}}_{p}$≈ 5.35 × 10^−4^ m^3^ mol^−1^. As a caution, the significant paper on water management by Berg et al. [33] gives *c*_f_ = 1200 mol m^−3^ which seems to be erroneous for Nafion 1100 if measured against the dry density. Even at high water content, which will lower overall density according to (10) below, this value seems too low.

### 2.3. Essential Transport Phenomena in the Membrane

Membrane transport phenomena must be described in a PEMFC model to account for the balance across the membrane of the observable quantities of interest (shown schematically in Figure 2) [13]. Spatial variation in these quantities may be of interest: both through the plane of the membrane from anode to cathode, and also in the plane of the membrane, in the case of 2D/3D models that capture spatial variations in the electrode plane due to flow channel design.

For an electrochemical device, the most common lumped quantity of interest is the electrochemical voltage, with the loss of cell voltage due to the membrane corresponding primarily to resistance to the transport of charge. Since charge is transported in the membrane in the form of protons (or, proton-carrying species) it is accompanied by electroosmotic flow of water; thus it is normally necessary for transport of the following conserved properties to be considered:charge;proton mass;water mass.

In the PEMFC context, it is, therefore, normally considered *essential* to define transport relations for:proton flux (current density);water flux.

Section 3 below will consider simpler models where the water transport is considered ideal, so that the membrane is uniformly hydrated and a charge transport-only model can be used. Section 4 and Section 5 will then consider the quantitative theory of water uptake and describe various models coupling current density to water flux.

If required, the model may be extended by the consideration of other transport phenomena:momentum (flow/mechanical stress, discussed in Section 7);heat (discussed in Section 9);dilute dissolved gas mass, to account for gas crossover (discussed in Section 10.1);dissolved ion mass (other than protons, discussed in Section 10.2).

## 3. Charge Transport-Only Membrane Models

### 3.1. Zero-Dimensional (0D, Lumped) Resistance Models

The simplest level of description of the electrochemical performance of a PEMFC is an empirical fit to the electrochemical performance as evidenced by a measured polarisation curve, without any physical resolution of the underlying phenomena. Ignoring transport phenomena, a simple fit resolving the kinetic and ohmic regions of the polarisation curve (cell voltage *E*_cell_ as a function of cell current density *i*_cell_) is [34]:(2)$$E_{\mathrm{cell}}=E_{\mathrm{OCV}}-A_{\mathrm{cat}}\log_{10}\left(\frac{i_{\mathrm{cell}}}{i_{\mathrm{ref}}}\right)-R_{\Omega }i_{\mathrm{cell}}$$

The parameters in this fit are the open circuit voltage *E*_OCV_, Tafel slope *A*_cat_, reference current density *i*_ref_, and ohmic series resistance *R*_Ω_ (Ω·m^2^). These parameters are determined empirically from the polarisation curve data. *R*_Ω_ is traditionally attributed primarily to the finite proton conductivity of the membrane (*κ*). Thus, for a membrane of thickness *L*_mem_:(3)$$R_{\Omega }\approx \frac{L_{\mathrm{mem}}}{\kappa }$$

For the thinnest membranes this approximation is less reliable, since the contributions from proton transport in the CL and from electrical contact resistances in the cell may become proportionally significant.

### 3.2. Constant Hydration Models

In uncontaminated operating conditions for a PEMFC, there are no dissolved ions in the membrane other than protons: hence, only protons contribute to *mobile* charge in the membrane, with the counter-ions present as the static sulfonate groups. Under these conditions, charge transport and proton mass transport are equivalent phenomena—neither one may take place without the other, and so the parameterisation of the transport of the two properties is inextricable. The current density **i** and molar flux of protons **N**_+_ differ only by means of a scaling by the Faraday constant *F*:(4)$$\frac{i}{F}=N_{+}$$

Although water balance is most often included, if a constant hydration condition is assumed then a charge transport-only transport theory results [35,36]. The simplest conductivity model is an Ohm’s law model of the current density (5) relating current density to proton conductivity (*κ*) and membrane-phase electrolyte potential (*φ*). This can then be combined with a statement of conservation of current in the bulk membrane, (6).
(5)i=−κ∇ϕ
(6)∇⋅i=0

Within a volume-averaged continuum model [37] of the CL, an effective conductivity may be used according to the volume fraction and connectivity of the ionomer within the CL composite [8,38]; also, the electrolyte current balance in the volume-averaged CL will have a source term corresponding to the faradaic current density (and, in principle, capacitive current density) and the corresponding source/sink of protons to the ionomer.

## 4. Sorption of Water

In a humid or wet environment, the membrane material takes up water through sorption. The water content *λ* of the polymer is defined as the ratio of moles of sorbed H_2_O (*n*_H_2_O_) to moles of sulfonic acid groups (*n*_SO_3__), within a defined reference volume of membrane (*V*) [1]:(7)$$\lambda =\frac{n_{H_{2}O}}{n_{{\mathrm{SO}}_{3}}}$$

The water content may equivalently be written in terms of a total mass of water (*m*_w_) taken up in the same reference volume:(8)$$\lambda =\frac{m_{w}}{M_{w}c_{f}V}$$
where the molar mass of water *M*_w_ = 0.018 kg mol^−1^.

By rearranging (7) and (8), the volume fraction of water in the hydrated PEM ($\phi_{w}$) is given:(9)$$\phi_{w}=\frac{\lambda }{\lambda +\frac{{\bar{V}}_{p}}{{\bar{V}}_{w}}}$$
where ${\bar{V}}_{w}$ is the molar volume of sorbed water ≈ 1.8 × 10^−5^ m^3^ mol^−1^ at 25 °C (corresponding to a density of sorbed water *ρ*_w_ ≈ 1000 kg m^−3^). The total density of hydrated polymer (*ρ*) approximately obeys a linear relation [1]:(10)$$\rho =\rho_{\mathrm{dry}}\left(1-\phi_{w}\right)+\rho_{w}\phi_{w}$$

From (9),
(11)$$\rho =\frac{M_{\mathrm{EW}}+M_{w}\lambda }{c_{f}^{-1}+\lambda {\bar{V}}_{w}}$$

We define a hygroscopic swelling coefficient *β*_w_ to account for membrane volume change under water uptake:(12)$$\beta_{w}\equiv \frac{V}{V_{\mathrm{dry}}}$$
(13)$$c_{w}=\frac{c_{f}}{\beta_{w}}\lambda$$

Springer et al. proposed the following linear correlation for *β*_w_ [39]:(14)$$\beta_{w}\approx 1+0.0126\lambda$$

### 4.1. Sorption Isotherms

A sorption isotherm relates the equilibrium water content *λ*_eq_ to the activity of water *a*_w_ in the membrane phase:(15)$$\lambda_{\mathrm{eq}}=\lambda_{\mathrm{eq}}\left(a_{w}\right)$$

To reach equilibrium, water may be sorbed from or desorbed to an adjacent phase, which might be either a gas phase containing water vapour at a certain activity, or a liquid phase (aqueous phase). These two cases are referred to as vapour-equilibrated (VE) and liquid-equilibrated (LE) conditions, respectively. It has been widely observed experimentally that the sorbed water uptake to PFSA ionomers is different between VE exposure to saturated water vapour and LE exposure to pure liquid water. Since both saturated water vapour and pure liquid water both have an activity *a*_w_ = 1, this result is thermodynamically unexpected and is often called “Schröder’s paradox” [40]: it is generally explained according to the presence of liquid water promoting an otherwise restricted phase transition of the ionomer that either eliminates the vapour–liquid interface near the membrane surface, or alters the energetics of the ionomer matrix–liquid contact [41,42,43,44].

Some experimental studies (notably Jeck et al. [45]) report an absence of Schröder’s paradox, but also suggest VE water contents that are higher than presented in other studies and closer to a typical LE value, possibly implicating the presence of a thin water film. The capability of thin water films to maintain bulk LE conditions in membranes with one VE face has also been suggested, based on X-ray tomography evidence [46].

The presence of VE or LE conditions also influences the interfacial resistance to the recovery of the sorption equilibrium, which will be discussed further below (Section 6) in the context of interfacial phenomena. It is relevant to note that whereas a PEMFC may be operated under VE conditions at both electrodes, PEMWE operation is likely to be LE at both electrodes, except possibly at high current density where gas production rate may reduce the extent of wetting. Likewise, liquid-fed DMFCs and similar devices would typically be liquid-equilibrated at the anode face of the membrane in contact with aqueous solution; in this case, however, the activity of water in the liquid phase is not necessarily equal to unity, due to the presence of the concentrated alcohol component.

Under VE conditions, the activity of water in the membrane (*a*_w_) can be specified as equal to the activity of water vapour in the equilibrating gas phase (*a*_w,vap_):(16)$$a_{w}=a_{w,\mathrm{vap}}\;(\mathrm{equilibrium})$$

The activity of water in the vapour phase can in turn be approximated as a function of the partial pressure of water (*p*_w,vap_) and the saturated partial pressure of water (vapour pressure, *p*_sat_) as a function of temperature (*T*):(17)$$a_{w,\mathrm{vap}}\approx \frac{p_{w,\mathrm{vap}}}{p_{\mathrm{sat}}(T)}$$
Equation (17) neglects fugacity corrections, which is normally suitable for PEMFC operating conditions at absolute pressures of a few bar.

The vapour pressure of water used in (17) is conventionally expressed as an empirical function of temperature [47]. Springer et al. used curve fitting on tabulated values for the vapour pressure to give the following standard expression, used also in recent PEMFC models (coefficient data tabulated in Table 1) [39,48]:(18)$$\log_{10}\left(\frac{p_{\mathrm{sat}}}{p_{0}}\right)=a_{0}+a_{1}\left(T-T_{0}\right)+a_{2}{\left(T-T_{0}\right)}^{2}+a_{3}{\left(T-T_{0}\right)}^{3}$$

Gurau et al. reported an alternative fit given by the American Society of Heating, Refrigerating and Air-Conditioning Engineers (ASHRAE) (coefficient data tabulated in Table 2) [49]:(19)$$\ln\left(\frac{p_{\mathrm{sat}}}{1\mathrm{Pa}}\right)=\frac{b_{-1}}{\left(T/1K\right)}+b_{0}+b_{1}\left(T/1K\right)+b_{2}{\left(T/1K\right)}^{2}+b_{3}{\left(T/1K\right)}^{3}+b_{e}\ln\left(T/1K\right)$$

The specification of the sorbed water activity under LE conditions typically depends on extending the range of values *a*_w_ to *a*_w_ > 1, according to a saturation-dependent definition of activity in the presence of liquid water. Simultaneously, the sorption isotherm (15) is extended to the corresponding values of *a*_w_. For instance, Springer et al. defined for the purposes of the sorption equilibrium that [39]:(20)$$a_{w}\equiv \frac{x_{w}p}{p_{\mathrm{sat}}}$$
where the mole fraction of water *x*_w_ includes both liquid- and gas-phase water. It should be noted that this definition is dependent on the use of a pseudo-two phase description of material transport in the GDL and CL, whereby liquid water is treated a gas-phase species with no independent momentum conservation. More detailed two-phase models, in which liquid water saturation in the porous diffusion media is described through an additional variable with a corresponding transport equation [10,50], may require an alternative specification of the LE sorption condition.

### 4.2. Empirical Sorption Models for Nafion 1100

A number of empirical sorption isotherms have been established experimentally based on measurements on Nafion 1100 [39,51,52,53,54]. All sorption isotherms reported in this subsection were parameterised for this material and, therefore, applications to other related materials—even in the Nafion family—should be undertaken with caution. A selection of the isotherms reported in this subsection are summarised in Figure 3.

The most commonly used sorption isotherm is an empirical polynomial relation due to Springer et al. [39]:(21)$$\begin{array}{l} \lambda_{\mathrm{eq}}=0.043+17.81a_{w}-39.85a_{w}^{2}+36.0a_{w}^{3},\;0\leq a_{w}\leq 1\\ \lambda_{\mathrm{eq}}=14+1.4\left(a_{w}-1\right),\;1\leq a_{w}\leq 3 \end{array}$$

The definition for supersaturated conditions (*a*_w_ > 1) aims to account for Schröder’s paradox through the definition (20) given above. The polynomial relation is specific to measurements at 30 °C, although the definition for *a*_w_ > 1 derives from liquid-equilibration data at 80 °C.

Kusoglu and Weber offered an alternative polynomial fit for VE conditions at 30 °C by averaging a wide range of experimental data (from different authors) [1]:(22)$$\lambda_{\mathrm{eq}}=0.05+20.45a_{w}-42.8a_{w}^{2}+36.0a_{w}^{3},\;0\leq a_{w}\leq 1$$

In spite of a variety of experimental investigations, there exist no definitive data for the temperature dependence of the sorption equilibrium at the higher temperatures more typical of PEMFC operation. Most data, however, suggest a relatively weak temperature dependence up to *T* = 90 °C: for this reason, it is common in the literature to encounter (21) being used at a typical PEMFC operating temperature also.

Hinatsu et al. measured the following sorption isotherm at 80 °C [55]:(23)$$\lambda_{\mathrm{eq}}=0.3+10.8a_{w}-16.0a_{w}^{2}+14.1a_{w}^{3},\;0\leq a_{w}\leq 1$$
while Pasaogullari et al. [56] presented a fit to data from Zawodzinski et al. [53] under VE conditions at 80 °C as:(24)$$\lambda_{\mathrm{eq}}=1.4089+11.263a_{w}-18.768a_{w}^{2}+16.209a_{w}^{3},\;0\leq a_{w}\leq 1$$

Kulikovsky extended the above expression to a super-saturated or LE condition [57]:(25)$$\lambda_{\mathrm{eq}}=0.3+6a_{w}\left(1-\tanh\left(a_{w}-0.5\right)\right)+3.9\sqrt{a_{w}}\left(1+\tanh\left(\frac{a_{w}-0.89}{0.23}\right)\right)$$

As shown in Figure 3, the Pasaogullari and Kulikovsky isotherms at higher temperature predict overall lower water contents than the Springer isotherm at ambient temperature, especially close to *a*_w_ = 1.

Meier and Eigenberger proposed a 25 °C isotherm (expressing activity as a function of water content) which is compatible with LE conditions but without exceeding *a*_w_ = 1 [58]:(26)$$\begin{array}{c} a_{w}=0.01355\lambda_{\mathrm{eq}}+0.03\lambda_{\mathrm{eq}}^{2},\;\lambda \leq 2.5\\ a_{w}=1.435+0.0022\lambda_{\mathrm{eq}}-\frac{2.75}{\lambda_{\mathrm{eq}}}-0.13\ln\lambda_{\mathrm{eq}},\;\lambda >2.5 \end{array}$$

Similarly, Karpenko-Jereb et al. originated an alternative empirical sorption isotherm with a smoothed jump at 0.97 ≤ *a*_w_ ≤ 1 to represent Schröder’s paradox, wherein the VE sorption isotherm shows no temperature dependence, but a temperature dependence is incorporated in the LE part [59]:(27)$$\begin{array}{l} \lambda_{\mathrm{eq},V}=1.55+13.71a_{w}-24.37a_{w}^{2}+21.87a_{w}^{3}\\ \lambda_{\mathrm{eq},L}=41.83\frac{T}{T_{0}}-28.31\\ \lambda_{\mathrm{eq}}=\lambda_{\mathrm{eq},V},\;a_{w}<0.97\\ \lambda_{\mathrm{eq}}=\lambda_{\mathrm{eq},V}+\left(\lambda_{\mathrm{eq},L}-\lambda_{\mathrm{eq},V}\right)\left(\frac{a_{w}-0.97}{0.03}\right),\;0.97\leq a_{w}\leq 1 \end{array}$$
with *T*_0_ = 303.15 K.

Futerko and Hsing gave the temperature dependence of the LE water content with *T*_0_ = 273 K as [60]:(28)$$\lambda_{\mathrm{eq},L}=10+0.0184\left(T-T_{0}\right)-9.9\times 10^{-4}{\left(T-T_{0}\right)}^{2}$$

### 4.3. Detailed Sorption Models

A number of authors have attempted to construct sorption models based on more fundamental properties of the PEM, rather than from a purely empirical basis. Again, all parameterisation in this subsection is reported for Nafion 1100.

Futerko and Hsing applied a Flory–Huggins model to express the activity of the membrane under VE conditions as [60]:(29)$$\begin{array}{c} a_{w}=\left(1-\phi_{m}\right)\exp\left(\left(1-\frac{{\bar{V}}_{w}}{{\bar{V}}_{p}}\right)\phi_{m}+\chi \phi_{m}^{2}\right)\\ \phi_{m}=\frac{{\bar{V}}_{p}+{\bar{V}}_{w}}{{\bar{V}}_{p}+\lambda {\bar{V}}_{w}} \end{array}$$
where the Flory parameter *χ* = 1.936 − (2.18 kJ mol^−1^)/*RT* and $\phi_{m}$ is the effective membrane volume fraction.

Thampan et al. suggested following the Brunauer–Emmett–Teller (BET) adsorption isotherm, expressed as follows [30]:(30)$$\lambda_{\mathrm{eq}}=\lambda_{\mathrm{eq},\mathrm{Thampan}}=\lambda_{\mathrm{mono}}\frac{\left(K_{1}\frac{a_{w}}{1-a_{w}}\right)\left(1-\left(n_{w,\mathrm{sat}}+1\right)a_{w}^{n_{w,\mathrm{sat}}}+n_{w,\mathrm{sat}}a_{w}^{1+n_{w,\mathrm{sat}}}\right)}{1+\left(K_{1}-1\right)a_{w}-K_{1}a_{w}^{1+n_{w,\mathrm{sat}}}}$$

Here, *λ*_mono_ is the water content corresponding to an effective ‘monolayer coverage’ within the polymer scaffold of the membrane, which was assumed = 1.8. The other parameters are fitting coefficients to experimental data [52,61] with values given as *K*_1_ = 150 and *n*_w,sat_ = 13.5. The BET isotherm predicts a saturated VE water content in terms of its parameters as [62]:(31)$$\lambda_{\mathrm{sat}}=\underset{a_{w}\to 1}{\lim}\lambda_{\mathrm{eq},\mathrm{Thampan}}=\lambda_{\mathrm{mono}}\frac{K_{l}n_{w,\mathrm{sat}}^{2}}{1+K_{l}n_{w,\mathrm{sat}}}$$

Klika et al. advocated the Guggenheim–Anderson–de Boer (GAB) isotherm, which extends the BET isotherm with an energy difference between the bulk and multilayer sorbed states of the water represented by the quantity *k*_G_, replacing *n*_w,sat_ in (30). Their equation is [63]:(32)$$\lambda_{\mathrm{eq}}=\lambda_{\mathrm{mono}}\frac{K_{1}k_{G}a_{w}}{\left(1-k_{G}a_{w}\right)\left(1+\left(K_{1}-1\right)k_{G}a_{w}\right)}$$
with empirical parameters fit to experimental data [39,45] as *λ*_mono_ = 1.93, *K*_1_ = 44.3 and *k*_G_ = 0.9.

Choi and Datta argued that Schröder’s paradox can be explained by the restricted morphology of the vapour–liquid interface within a pore, which is resolved by liquid equilibration [41]. Their work derived a rather complicated expression for the sorption isotherm that extends the Thampan isotherm (Equation (30)) for both vapour- and liquid-equilibrated modes, considering bound water as a separate Langmuirian contribution to the isotherm, with a BET model accounting for additional bound water uptake beyond a monolayer, and a Flory–Huggins isotherm for free water. The resulting, rather unwieldy implicit equations relate to (30) as:(33)$$\begin{array}{l} \left(1-\lambda_{\mathrm{eq},\mathrm{Thampan}}\right)\left(a_{w}\exp\left(\frac{{\bar{V}}_{w}}{RT}\left(\frac{\kappa_{\mathrm{sorp}}\lambda_{\mathrm{eq},L}}{\lambda_{\mathrm{eq},L}+\frac{{\bar{V}}_{p}}{{\bar{V}}_{w}}}\right)\right)-1\right)\lambda_{\mathrm{eq},L}=1\\ \left(1-\lambda_{\mathrm{eq},\mathrm{Thampan}}\right)\left(a_{w}\exp\left(\frac{{\bar{V}}_{w}}{RT}\left(\frac{\kappa_{\mathrm{sorp}}\lambda_{\mathrm{eq},V}}{\lambda_{\mathrm{eq},V}+\frac{{\bar{V}}_{p}}{{\bar{V}}_{w}}}-a_{\mathrm{pore}}\gamma_{w}\cos\theta_{c}\left(1+\frac{{\bar{V}}_{p}}{{\bar{V}}_{w}\lambda_{\mathrm{eq},V}}\right)\right)\right)-1\right)\lambda_{\mathrm{eq},V}=1 \end{array}$$

The fitted parameters were as above, except setting *K*_1_ = 100 and *n*_w,sat_ = 5. Additional parameters are defined in Table A2 in Appendix A, and given as *κ*_sorp_ = 183 atm, *a*_pore_ = 2.1 × 10^8^ m^−1^, *γ*_w_ = 0.0721 N m^−1^, *θ*_c_ = 98°.

In the context of a DMFC, Meyers and Newman developed an isotherm based on fundamental energetics of the PEM [64]. Like the model of Thampan et al. [30], this model combines an acid-base equilibrium for dissociation of sulfonic acid groups in the membrane with a requirement of electroneutrality. The resulting simultaneous equations that define the isotherm are (coefficient data tabulated in Table 3) [64]:(34)$$\begin{array}{l} a_{w}=K_{2}\left(\lambda_{\mathrm{eq}}-\lambda_{+}\right)\exp\left(\phi_{2}\lambda_{+}\right)\exp\left(\phi_{3}\lambda_{\mathrm{eq}}\right)\\ \lambda_{+}\exp\left(\phi_{1}\lambda_{+}\right)\exp\left(\phi_{2}\lambda_{\mathrm{eq}}\right)=K_{1}\left(1-\lambda_{+}\right)\left(\lambda_{\mathrm{eq}}-\lambda_{+}\right) \end{array}$$
where *λ*_+_ is the ratio of moles of hydronium ions to moles of sulfonic acid sites and must be solved for self-consistently. The φ*_n_* coefficients are defined as:(35)$$\begin{array}{c} \phi_{1}=\frac{2}{M_{\mathrm{EW}}}\left(E_{00}-2E_{++}-2E_{+0}\right)\\ \phi_{2}=\frac{2}{M_{\mathrm{EW}}}\left(E_{+0}-2E_{00}\right)\\ \phi_{3}=\frac{2}{M_{\mathrm{EW}}}E_{00} \end{array}$$

The Meyers–Newman isotherm was applied by Weber and Newman using the above parameters, subject to a further empirical modification to account for inaccuracies of the model at low water uptake [65]:(36)$$\lambda =\beta \lambda_{\mathrm{eq}}\left(1+\exp\left(\lambda_{\mathrm{eq},\mathrm{crit}}-\lambda_{\mathrm{eq}}\right)\right)$$
with the scaling coefficient *β* = 1 (included for generality, see Section 4.4, “Sorption within the Catalyst Layer” below) and *λ*_eq,crit_ = 0.3. Subsequently, the influence of temperature on this sorption isotherm was accounted for by treating all inputs as temperature-independent except *K*_2_, which varies as [66]:(37)$$K_{2}=0.217\;\exp\left(\frac{\Delta H_{\mathrm{sorp}}}{R}\left(\frac{1}{T_{0}}-\frac{1}{T}\right)\right)$$
with *T*_0_ = 303.15 K and the enthalpy change of sorption Δ*H*_sorp_ = +1 kJ mol^−1^.

Murahashi et al. [67] and Eikerling and Berg [44] and have both argued for the influence of size distribution of pores upon the sorption isotherm. Smaller pores may retain water due to the pressure drop of the liquid–vapour interface, even when larger pores become dehydrated [67]. Eikerling and Berg argued that pores with a range of surface charge densities can wet progressively due to more highly charged pores taking up water more slowly, but attaining larger limiting wetted radii [68]. Since the Eikerling–Berg description makes a number of highly specific assumptions about the geometry and governing phenomena of the matrix-liquid-vapour system, it could be viewed as didactic rather than being directly usable for quantitative modelling of membrane sorption in the PEMFC context.

### 4.4. Sorption within the Catalyst Layer

It has been established experimentally that membrane material within the composite structure of the catalyst layer takes up proportionally less water by sorption than in the bulk membrane [69,70]. This has been attributed to the altered internal morphology of Nafion ionomer when present as a <60 nm film. Recent analysis has indicated that a lamellar structure forms for particularly thin films, and that total water uptake varies non-monotonically with film thickness [71]. This work has also emphasised that for thin films, homogenisation of material properties is typically unreasonable, and interfacial phenomena may dominate.

It has been proposed that the CL water uptake could be described empirically by setting *K*_2_ = 0.231, *β* = 0.342 (all other parameters identical) in the Meyers–Weber–Newman isotherm (36) [70]. This work also assumed a constant apparent proton conductivity *κ*_eff_ = 10^−4^ S m^−1^ in the CL. Incorporation into a full PEMFC model was reported to give improved accuracy in the prediction of performance loss due to anode dehydration, under low relative humidity operating conditions. From a thermodynamic point-of-view, it is necessary to recognise that altering the isotherm in the CL specifically implies a phase transition for water between the CL ionomer and the bulk membrane, at constant activity. If no resistance is incorporated for this transfer, reduced water uptake in the CL remains compatible with normal water content and water transfer fluxes in the bulk membrane.

Additionally, studies have suggested that sorbed water uptake of the CL depends upon Pt loading [69], choice of carbon support [38,72], and the solvent used in CL preparation [73]. Using the Pt/C-phase effective electronic conductivity as a probe, Morris et al. showed that sorption/desorption in the CL appeared to be free from hysteresis [74].

Mashio et al. provided a comprehensive model for CL water uptake in which sorption to the membrane was incorporated by means of a Langmuir isotherm [72]:(38)$$\lambda =\lambda_{\mathrm{sat}}\frac{K_{\mathrm{mem}}a_{w}p_{\mathrm{sat}}}{1+K_{\mathrm{mem}}a_{w}p_{\mathrm{sat}}}$$
with equilibrium constant *K*_mem_ = 3.3 × 10^−4^ Pa^−1^; this work stresses the role of water adsorption on a variety of materials within the CL, in terms of overall water uptake of this region.

Since the CL has a high volumetric surface area of contact between vapour- or liquid-phase water and the membrane material, the local sorption properties in this region may significantly influence the overall water balance of the membrane. Kosakian et al. have recently presented a dedicated CL isotherm as follows [75]:(39)$$\begin{array}{l} \lambda_{\mathrm{eq}}=\left(6.932a_{w}-14.53a_{w}^{2}+11.82a_{w}^{3}\right)\exp\left(\theta_{\mathrm{sorp}}\left(\frac{1}{T_{0}}-\frac{1}{T}\right)\right),\;0\leq a_{w}\leq 1\\ \lambda_{\mathrm{eq}}=22,\;a_{w}>1 \end{array}$$
with *θ*_sorp_ = 2509 K, *T*_0_ = 303.15 K. It should be noted that this specification gives a large discontinuity at the transition to liquid equilibration at *a*_w_ = 1; this is not depicted in Figure 3 which focuses on VE conditions.

## 5. Coupled Proton-Water Transport

As introduced through the discussion above in Section 2.3 (“Essential Transport Phenomena in the Membrane”), current flow by proton flux through the membrane is always accompanied by water transport due to electroosmotic drag. Therefore, the majority of practical membrane models used in PEMFC simulation are coupled models incorporating both proton and water transport. In this section, we will introduce some essential considerations surrounding the coupling of the two transport processes, and then consider three principal approaches to this coupling and their parameterisation.

First, we will discuss the membrane model originated in the seminal early PEMFC simulation work by Springer, Zawodzinski and Gottesfeld [39] (hereafter “Springer model” for brevity). This widely used model accounts for electroosmotic drag and water diffusion in an essentially empirical manner. Second, the Weber–Newman model [65] will be considered; this model is rooted in a more formal derivation from non-equilibrium thermodynamics of the membrane phase, but remains empirically parameterised. Lastly, the binary friction model developed by Djilali and co-workers will be discussed [76,77,78].

Weber and Newman have advocated the interpretation of the hydrated membrane as a system with three chemical components: water, protons, and fixed membrane structure [65]. Therefore, there exist no more than three independent transport properties associated with the binary interactions of the three components. If bulk momentum transfer through the membrane is considered, mechanical resistance (friction) may account for a fourth transport property. The three most experimentally accessible choices for the definition of the three independent transport properties, following the Weber–Newman scheme, are [65]:proton conductivity *κ*—the ratio of current density to electrolyte potential gradient for uniform water content;electroosmotic drag coefficient *ξ*—the ratio of water flux to current density for uniform water content;water diffusivity *D*_w_—the ratio of water flux to water concentration gradient for zero current density.

Parameterisation for each of these intrinsic properties (proton conductivity, water diffusivity, electroosmotic drag coefficient) will be summarised in Section 5.4, Section 5.5, Section 5.6. The rigorous measurement of these properties for Nafion 1100 was initiated in the early 1990s in a series of works by Zawodzinski et al. [52,53,79,80] and Fuller and Newman [81].

Since the fixed membrane structure is considered rigid (there is no mechanical translation of the membrane), the Weber–Newman scheme considers only water and proton fluxes, and each of these fluxes has a conjugate thermodynamic variable whose gradient indicates the direction and magnitude of the flux. Formal treatments define [65,82]:membrane-phase electrolyte potential φ as the thermodynamic variable conjugate to the driving force for current flow, under uniform hydration;chemical potential of water *μ*_w_ (expressed as required in terms of the local water content *λ*) as the thermodynamic variable conjugate to the driving force for water flux, at zero proton current.

From a thermodynamic standpoint, the inclusion of pressure as a local variable, in addition to electrolyte potential and water content, is almost certainly an overdetermination of the system except for the liquid-equilibrated case; it has not found general support [65,78]. Exceptions in the liquid-equilibrated case, where free water channels may be present, will be discussed further in Section 5.7 below.

### 5.1. Springer Membrane Model

Springer et al. defined the flux of water through the membrane (**N**_w_) phenomenologically as the sum of a Fickian diffusion term in water concentration *c*_w_, and an electroosmotic drag term [9,39]:(40)$$N_{w}=-D_{w}\nabla c_{w}+\xi \frac{i}{F}$$

This is compatible with the transport property definitions given above; non-linear transport behaviour is implied if the coefficients *D*_w_ and *ξ* are themselves functions of water content. Using (14):(41)$$N_{w}=-D_{w}c_{f}\nabla \left(\frac{\lambda }{\beta_{w}}\right)+\xi \frac{i}{F}$$

On the basis that *β*_w_ is a function of water content only, the Springer water flux formula can be further abbreviated. Springer et al. expressed the water gradient $\nabla_{\mathrm{dry}}$ with respect to a fixed, dry-membrane coordinate which is undeformed by swelling under water uptake [39]:(42)$$N_{w}=-D_{\lambda }c_{f}\nabla_{\mathrm{dry}}\lambda +\xi \frac{i}{F}$$
where the apparent diffusion coefficient with respect to water content (*D_λ_*) is given:(43)$$D_{\lambda }=\frac{D_{w}}{\beta_{w}^{2}}\left(1-\frac{\lambda }{\beta_{w}}\frac{\partial \beta_{w}}{\partial \lambda }\right)$$

While the original Springer et al. work considered membrane expansion under swelling, subsequent works have assumed that the compressed membrane has no thickness variation [83]; the role of compression is discussed further below in Section 7.2. Thus one can write simply:(44)$$N_{w}=-D_{\lambda }c_{f}\nabla \lambda +\xi \frac{i}{F}$$

Typically, this diffusivity *D_λ_* is parameterised directly from experimental data, as discussed further in Section 5.5 below. Within the membrane, conservation of water mass requires that, under steady-state conditions:(45)$$\nabla \cdot N_{w}=0$$

The widely used Springer model describes coupled proton-water transport in PEMFC membranes by combining Equations (44) and (45) with the Ohm’s law expressions (5) and (6). The interaction of the two transport processes is expressed by the electroosmotic drag term in (44) and the water content-dependence of the proton conductivity in (5).

### 5.2. Weber–Newman Membrane Model

The Weber–Newman model is an alternative to the Springer model that seeks to consider the non-equilibrium thermodynamics of the coupled proton and water transport processes in a more formal and consistent manner [65]. Moreover, the empirically determined transport properties are assumed to have different definitions in the VE and LE regimes.

The substantive difference between the transport equations used for the Weber–Newman model and the Springer model is the presence of a cross-term contribution to the current density expression due to water streaming current, which is non-zero wherever the water chemical potential is non-uniform through the membrane. This phenomenon accounts for the symmetry of the binary proton–water interaction: just as electroosmotic drag describes the motion of water molecules due to proton current, so the streaming current describes the motion of protons due to water diffusion. Specifically:(46)$$i=-\kappa \nabla \phi -\frac{\kappa \xi }{F}\nabla \mu_{w}$$

The water flux is then given as, alongside (46) [65]:(47)$$N_{w,m}=-\alpha_{w,m}\nabla \mu_{w,m}+\frac{\xi_{m}}{F}i$$
where the subscript *m* = V or L and indicates VE or LE conditions.

The relation between the intrinsic transport coefficient *α*_w_ and the apparent Fick’s law diffusion coefficient *D_λ_* has been expressed differently by various authors [65,83]. A simple interpretation for the VE case has been given by Setzler and Fuller, on the basis that the water content is the only parameterising variable for the local state of the hydrated membrane:(48)$$\begin{array}{ccc} -\alpha_{w,V}\nabla \mu_{w} & = & -D_{\lambda }c_{f}\nabla \lambda \\ & = & -D_{\lambda }c_{f}\frac{\partial \lambda }{\partial a_{w}}\frac{\partial a_{w}}{\partial \mu_{w}}\nabla \mu_{w}\\ & = & -D_{\lambda }c_{f}\frac{\partial \lambda }{\partial a_{w}}\frac{a_{w}}{RT}\nabla \mu_{w}\\ & = & -D_{\lambda }\frac{c_{f}\lambda }{RT}\frac{\partial \ln\lambda }{\partial \ln a_{w}}\nabla \mu_{w}\\ & = & -D_{\lambda }\frac{c_{w}}{RT}\frac{\partial \ln\lambda }{\partial \ln a_{w}}\nabla \mu_{w} \end{array}$$

If the self-diffusion coefficient *D_μ_* is defined as given by Springer et al. [39]:(49)$$\alpha_{w,V}\equiv D_{\mu }\frac{c_{w}}{RT}$$
then the two diffusion coefficients are simply related by the Darken factor [83]:(50)$$D_{\lambda }=\frac{\partial \ln a_{w}}{\partial \ln\lambda }D_{\mu }$$

However, it should be noted that in their original work, Weber and Newman used instead of the definition (49) the following definition based on the Maxwell–Stefan diffusion coefficient for the membrane-water system as a binary system [65]:(51)$$\alpha_{w,V}\equiv D_{\mu ,\mathrm{WN}}\frac{c_{w}}{RT}\left(1+\lambda \right)$$

In interpreting the diffusivity data given below in Section 5.5, the inequivalence of (49) and (51) must be borne in mind.

### 5.3. Binary Friction Model (BFM)

The binary friction model (BFM) is derived beginning from a generalised diffusion equation presented succinctly in the following form [82]:(52)$${\bar{V}}_{w}\left(\begin{array}{c} N_{+}\\ N_{w} \end{array}\right)=-\left(\begin{array}{cc} D_{11} & D_{12}\\ D_{21} & D_{22} \end{array}\right)\left(\begin{array}{c} \frac{F}{RT}\nabla \phi \\ \nabla \lambda \end{array}\right)$$
where the generalised diffusion coefficients *D_mn_* are functions of *λ*,*T*, thereby accounting for non-linearity.

In the BFM developed by Fimrite, Carnes, Struchtrup and Djilali [76,77], the mole fraction of proton carriers is derived using a method originated by Thampan et al. [30] This mole fraction is then applied as a dependent variable in a concentrated solution theory; Shah et al. have also applied the Thampan method previously to dilute solution theory [84]. The Thampan approach assumes that there exists an acid-base equilibrium quantifying the degree of dissociation of the sulfonic acid groups in the fixed membrane structure [30]:(53)$${\mathrm{SO}}_{3}H(\mathrm{fixed})+H_{2}O\overset{K_{a,\mathrm{mem}}}{\underset{}{⇄}}{\mathrm{SO}}_{3}^{-}(\mathrm{fixed})+H_{3}O^{+}$$

In this case, the total water content is divided between neutral water and charged hydronium species. Electroneutrality requires that the local concentrations of H_3_O^+^ and SO_3_^−^ are equal. Later iterations of the BFM employed the simplifying assumption that sulfonic acid dissociation is complete (strong acid behaviour) [78,82].

In the original BFM, there were five fitted parameters, but no further empirical relations to water content [77]. Fimrite et al. thereafter used the term “BFM2” for a specialised binary friction model specific to the PEMFC device context, in the limit of complete sulfonic acid group dissociation [78,82]. This model has six parameters, which were fitted to a conductivity vs. water content curve derived from alternating current (AC) impedance measurements. (As an aside, the original derivation of the BFM2 model is complicated by two irregular negative multiples: one in the definition of the potential driving force ([82], eqn 3) and then one in the charge of the hydronium ion (set = −1 in [82]). These negatives cancel in the derived equations.) The resulting Weber–Newman transport coefficients are then expressed in terms of a set of binary diffusivities as [82]:(54)$$\kappa =f_{\mathrm{eff}}c_{f}\frac{F^{2}}{RT}\frac{D_{+m}}{\lambda }\frac{D_{0m}+D_{+0}\lambda }{D_{+0}\lambda +D_{+m}+D_{0m}\left(\lambda -1\right)}$$
(55)$$\xi =\frac{D_{0m}\left(\lambda -1\right)}{D_{0m}+D_{+0}\lambda }$$
(56)$$D_{w}=f_{\mathrm{eff}}\frac{D_{0m}}{\lambda }\frac{D_{+0}}{D_{+0}\lambda +D_{+m}+D_{0m}\left(\lambda -1\right)}$$

The three binary diffusivities are expanded further as functions of temperature and/or water content, as:(57)$$D_{km}=D_{+0}A_{k}\lambda^{s}$$
(58)$$D_{+0}=D_{+0,\mathrm{ref}}\exp\left(\theta_{\mathrm{diff}}\left(\frac{1}{T_{\mathrm{ref}}}-\frac{1}{T}\right)\right)$$
and the coefficient *f*_eff_ accounts for the influence of an effective ‘membrane porosity’ by the relation:(59)$$f_{\mathrm{eff}}={\left(\phi_{w}\left(\lambda \right)-\phi_{w}\left(\lambda_{\min}\right)\right)}^{q}$$

The more explicit definition of the functional forms of each of the three orthogonally measurable transport coefficients in terms of the various water-content-independent parameters is suggested as a means to reduce the degree of empiricism in the model setup (coefficients are tabulated in Table 4).

Djilali and Sui [85] have argued that the above theory offers improved sensitivity to the variation in membrane behaviour under conditions of low anode humidification, where the empirical correlations from Springer et al. [39] may be less reliable. Conversely, it is unclear whether the BFM will retain applicability in the limit of high water content or liquid-equilibration, where hydraulic transport becomes significant.

### 5.4. Proton Conductivity as a Function of Water Content

The proton conductivity *κ* appears in both the Springer model and the Weber–Newman model as an empirical function of water content. Within the Weber–Newman model, proton conductivity is additionally considered to have a different value under LE conditions. At the atomic scale, proton transport in a PEM is understood to occur due to two possible mechanisms: the vehicle mechanism, in which protons are carried in the form of hydronium ions, H_3_O^+^; and the hopping (Grotthuss) mechanism, in which protons are transported by a hydrogen bond-mediated long-range rearrangement of the bonding network between water molecules and hydronium ions, as in bulk liquid water [42]. Since membranes at lower hydration do not contain connected liquid regions with extensive hydrogen bonding networks, the Grotthuss mechanism is suppressed and the vehicle mechanism is believed to dominate [1,2]. Activation of the Grotthuss mechanism for higher water content, especially when liquid-equilibrated, increases the conductivity as water content rises.

Various empirical relations reported in this subsection for proton conductivity of Nafion 1100 materials are plotted in Figure 4 (*T* = 30 °C) and Figure 5 (*T* = 80 °C).

The functional dependence of conductivity with respect to water content has been expressed in terms of a polynomial relationship, where there exists some minimum water content *λ*_0_ required for proton conductivity, based on a percolation theory in which there is a need to form connected hydrated channels through which protons can migrate [42,86]. Additionally, an activation energy is included for non-isothermal processes.
(60)$$\begin{array}{ccc} \kappa & = & 0,\;\lambda <\lambda_{0}\\ & = & \kappa_{0}{\left(\lambda -\lambda_{0}\right)}^{n_{\mathrm{cond}}}\exp\left(\theta_{\mathrm{cond}}\left(\frac{1}{T_{0}}-\frac{1}{T}\right)\right),\;\lambda \geq \lambda_{0} \end{array}$$

For Nafion 1100, the average value for the exponent *n*_cond_ through a range of studies [1,59] is ≈ 1.5, but some empirical models have applied other data values, as summarised in Table 5. In particular, early studies on proton conductivity as a function of water content supported an approximately linear relation [39,52]. Relations for which *θ*_cond_ is undefined apply only at *T* = *T*_0_.

The lower activation energy (*θ*_cond_ ≈ 1300 K) was supported by the experimental measurements by Karpenko-Jereb et al. [59]. Moreover, this work sets the dependence strictly in terms of volume fraction rather than water content:(61)$$\begin{array}{ccc} \kappa & = & 0,\;\phi_{w}<\phi_{w,0}\\ & = & \kappa_{0,\phi }{\left(\phi_{w}-\phi_{w,0}\right)}^{n_{\mathrm{cond}}}\exp\left(\theta_{\mathrm{cond}}\left(\frac{1}{T_{0}}-\frac{1}{T}\right)\right),\;\phi_{w}\geq \phi_{w,0} \end{array}$$

This gives approximate equivalence to the tabulated values above when the denominator in (13) is assumed to be constant.

Ju et al. applied the Springer data to a GORE-SELECT membrane with a scaling factor 0.5 to account for tortuosity due to the reinforcement [89].

Ramousse et al. repeated a higher-order polynomial fit due to Neubrand [90,91]:(62)$$\kappa /Sm^{-1}=\left(0.2658\lambda +0.0298\lambda^{2}+0.0013\lambda^{3}\right)\exp\left(\theta_{\mathrm{cond}}\left(\frac{1}{T_{0}}-\frac{1}{T}\right)\right)$$
with a water content-dependent activation energy:(63)$$\theta_{\mathrm{cond}}/K=2640\exp\left(-0.6\lambda \right)+1183$$

Dobson et al. quoted a polynomial expression for Nafion 211 [92]:(64)$$\kappa /Sm^{-1}=\left(2.0634+1.052\lambda +0.010125\lambda^{2}\right)\exp\left(\theta_{\mathrm{cond}}\left(\frac{1}{T_{0}}-\frac{1}{T}\right)\right)$$
with *θ*_cond_ = 751.4 K.

Using the wide data set for proton conductivity compiled by Sone et al. [93], Baschuk and Li tabulated third-order polynomial fit data for the water-content dependence of conductivity, at a range of temperatures from *T* = 20 °C to *T* = 70 °C [94].

Setzler and Fuller used impedance measurements under varying relative humidity conditions to produce the following empirical relation for Nafion 212 at 80 °C [95]:(65)$$\kappa =\kappa_{0}\exp\left(\alpha_{\lambda }\frac{\lambda }{\lambda_{\mathrm{crit}}}\right)$$
with *κ*_0_ = 1.55 S m^−1^ and *α_λ_* = 2.2.

Weber and Newman defined the conductivity under LE and VE conditions (indexed as *m* = L or *m* = V below) as depending upon the local volume fraction of water present under the given equilibration condition, without any specification of how the empirical expressions were derived [65]:(66)$$\kappa_{m}/Sm^{-1}=\{\begin{array}{cc} 10^{-9} & \phi_{w,m}<\phi_{w,\mathrm{crit}}\\ \kappa_{0}{\left(\phi_{w,m}-\phi_{w,\mathrm{crit}}\right)}^{\frac{3}{2}}\exp\left(\frac{E_{A,\mathrm{cond}}}{R}\left(\frac{1}{T_{0}}-\frac{1}{T}\right)\right) & \phi_{w,\mathrm{crit}}\leq \phi_{w,m}\leq \phi_{w,\max}\\ \kappa_{0}{\left(\phi_{w,\max}-\phi_{w,\mathrm{crit}}\right)}^{\frac{3}{2}}\exp\left(\frac{E_{A,\mathrm{cond}}}{R}\left(\frac{1}{T_{0}}-\frac{1}{T}\right)\right) & \phi_{w,m}>\phi_{w,\max} \end{array}$$
with *κ*_0_ = 50 S m^−1^ at *T*_0_ = 298.15 K, *E*_A,cond_ = +15 kJ mol^−1^, and φ_w,crit_ = 0.06, φ_w,max_ = 0.45.

Kosakian et al. have presented a specific formulation for proton conductivity in the CL ionomer (below) [75]. It should be noted that this expression has no stated correlation to porosity or tortuosity properties of the ionomer in the CL composite, so the absolute magnitude of the conductivity it reports cannot be applied generally to any CL; however, the functional form of the water content-dependence might be considered more widely applicable.
(67)$$\kappa =\left(\overset{3}{\underset{i=0}{\sum }}a_{i}\omega^{i}\right)\exp\left(\frac{E_{A,\mathrm{cond}}}{R}\left(\frac{1}{T_{0}}-\frac{1}{T}\right)\right)$$
(68)$$\omega =\{\begin{array}{cc} 100\left(\left(\overset{3}{\underset{i=0}{\sum }}b_{i}\lambda^{i}\right)\right) & 0<\lambda <13\\ 100 & \lambda \geq 13 \end{array}$$
with *E*_A,cond_ = +15 kJ mol^−1^ (polynomial data tabulated in Table 6).

### 5.5. Water Diffusivity as a Function of Water Content

Implementations of the Springer model typically reference the water diffusivity measured by Zawodzinski et al. using nuclear magnetic resonance (NMR) methods with respect to chemical potential gradients [79], and then solve the sorption isotherm implicitly to convert to a diffusion coefficient with respect to water content. This Fickian diffusion approach has been criticised by Janssen in super-saturated (LE) conditions, because it depends on an extrapolation of the derivative of the sorption isotherm that is defined inexactly in the limit of super-saturation [96]. Eikerling et al. reported poor correlation of the Springer diffusion data with experimental measurements [97]; however, this appears to have been a comparison made in conjunction with proton conductivity data not matching those used in the Springer work.

The corresponding diffusion coefficient shows a characteristic peak in the range *λ* = 3 to 4. The original fit used by Springer et al. was reported incompletely [39] and is now deprecated, since it was later refined by Motupally et al. [83]:(69)$$D_{\lambda }/m^{2}s^{-1}=3.1\times 10^{-7}\left(\exp\left(0.28\lambda \right)-1\right)\exp\left(-\frac{\theta_{\mathrm{diff}}}{T}\right),\;\lambda <3$$
(70)$$D_{\lambda }/m^{2}s^{-1}=4.17\times 10^{-8}\left(1+161\exp\left(-\lambda \right)\right)\exp\left(-\frac{\theta_{\mathrm{diff}}}{T}\right),\;\lambda \geq 3$$
where *θ*_diff_ = 2436 K. The Motupally diffusivity model expressed by (69) and (70) was supported by water flux measurements by St-Pierre [98], and has been used in a number of subsequent modelling works.

The data recorded by Okada et al. suggested a constant value of *D*_w_ = 5 × 10^−10^ m^2^ s^−1^ [99] which has been used subsequently as *D_λ_* = 3 × 10^−10^ m^2^ s^−1^ [91]. In a recent work, Kosakian et al. argued from empirical evidence that their own data could be fit accurately by multiplying the Motupally terms by a multiple of 3.2 [75].

The following linear model was measured by Fuller and Newman [100]:(71)$$D_{\lambda }=D_{0,\lambda }\lambda \exp\left(-\frac{\theta_{\mathrm{diff}}}{T}\right)$$
with *D*_0,λ_ = 2.1 × 10^−7^ m^2^ s^−1^ (as converted by Motupally et al. [83]) and *θ*_diff_ = 2436 K. Karpenko-Jereb et al. reported a corresponding value *D*_0,*λ*_ = 7.84 × 10^−8^ m^2^ s^−1^ with *θ*_diff_ = 2383 K [59].

Alternative fits to the data of Springer et al. and Zawodzinski et al. [39,79] have been reported by subsequent authors. For example, Nguyen and White were guided by experimental observations in a PEMFC configuration to scale water diffusivity by the electroosmotic drag coefficient *ξ* according to the relation [101]:(72)$$D_{w}=D_{0,w}\xi \exp\left(-\frac{\theta_{\mathrm{diff}}}{T}\right)$$
with *D*_0,w_ = 1.6 × 10^−7^ m^2^ s^−1^ and *θ*_diff_ = 2416 K.

Mazumder reported an alternative fit as follows [102]:(73)$$D_{\lambda }/m^{2}s^{-1}=2.9\times 10^{-7}f(\lambda )\exp\left(-\frac{\theta_{\mathrm{diff}}}{T}\right)$$
(74)$$\begin{array}{l} f(\lambda )=1,\;\lambda \leq 2\\ f(\lambda )=1+2\left(\lambda -2\right),\;2<\lambda \leq 3\\ f\left(\lambda \right)=3-1.38\left(\lambda -3\right),\;3<\lambda \leq 4\\ f\left(\lambda \right)=2.563-0.33\lambda +0.0264\lambda^{2}-0.000671\lambda^{3},\;\lambda \geq 4 \end{array}$$

Gurau et al. applied the Mazumder polynomial fit (for *λ* ≥ 4) across the full water content range [49]. This water content dependence has also been applied to models of GORE-SELECT membranes, but with an altered pre-factor to give approximately half the diffusivity of pure Nafion 1100, due to tortuosity from the reinforcement [103].

Data recorded by van Bussel et al. [87] have been used by models developed by Kulikovsky [57] as well as Wu et al. [104]. These data lack any variation of membrane thickness and so may be perturbed by interfacial phenomena: they also lack the characteristic peak at low *λ* given by (69) and (70). The following fit is given by Kulikovsky, at *T* = 80 °C [57]:(75)$$D_{w}/m^{2}s^{-1}=4.1\times 10^{-10}{\left(\frac{\lambda }{25}\right)}^{0.15}\left(1+\tanh\left(\frac{\lambda -2.5}{1.4}\right)\right)$$

Figure 6 plots various correlations for diffusion coefficient at operating temperature, assuming in the case of (75) that *D*_w_ = *D_λ_* (that is, ignoring swelling corrections according to (43)). It should be noted that the various reported equations do not give close agreement, and differ by over an order of magnitude in the limit of high water content. One possible reason for this could be the unreliable extrapolation of data measured close to room temperature to much higher temperatures, but the extent of inconsistency merits further controlled measurements on contemporary state-of-the-art materials.

Weber and Newman parameterised the self-diffusion coefficient for Nafion 1100 as [65]:(76)$$D_{\mu ,\mathrm{WN}}=D_{\mu ,0}\phi_{w}\exp\left(-\theta_{\mathrm{diff}}\left(\frac{1}{T_{0}}-\frac{1}{T}\right)\right)$$
with *D_μ_*_,0_ = 1.8 × 10^−9^ m^2^ s^−1^, *T*_0_ = 303.15 K and *θ*_diff_ = 2400 K. For use in the CL specifically, Kosakian et al. gave correspondingly [75]:(77)$$D_{\mu }=D_{\mu ,0}\phi_{w}\exp\left(-\theta_{\mathrm{diff}}\left(\frac{1}{T_{0}}-\frac{1}{T}\right)\right)$$
with *D_μ_*_,0_ = 5.44 × 10^−9^ m^2^ s^−1^ and other parameters the same. Here, the substantial difference in magnitude of the pre-factor presumably arises from the different definitions of self-diffusion coefficient in each case (see above, Section 5.2).

### 5.6. Electroosmotic Drag Coefficient

Electroosmotic drag is known to increase with higher water content (liquid-equilibrated membrane), and with temperature under LE conditions; measurements under different conditions or with different experimental methods have yielded significant disparity [105]. The need for careful control of hydration conditions has been emphasised [106]; however, due to interfacial water transport resistances (see Section 6.2 below), electroosmotic drag itself may induce a water content gradient, even in configurations where the humidity on each face of the membrane is rigorously controlled. Therefore, care is always needed in interpreting experimental data. Also, the literature is often unclear due to the occasional use of “electroosmotic drag coefficient” to indicate the phenomenological property of net number of water molecules transferred per proton transferred, *K*_drag_:(78)$$K_{\mathrm{drag}}\equiv F\frac{\left|N_{w}\right|}{\left|i\right|}$$

This quantity should not be confused with the intrinsic transport property *ξ* appearing in the water transport Equations (42) and (47).

Some of the parameterisations reported in this subsection for the electroosmotic drag coefficient in Nafion 1100 are presented in Figure 7.

The original parameterisation of the Springer model used a linear relation in *λ*, tending to zero for zero water content [39]:(79)$$\xi =a_{\xi ,\lambda }\frac{\lambda }{\lambda_{\mathrm{sat}}}$$
with *a_ξ,λ_* = 2.5, *λ*_sat_ = 22 for Nafion 1100. This was supported by subsequent data from Okada et al. [99]. However, several later studies have presented evidence that the electroosmotic drag coefficient is constant and very close to unity up to a critical water content associated with the liquid-equilibration phase transition, whereafter it rises linearly (data tabulated in Table 7) [80,87,103,107]:(80)$$\begin{array}{l} \xi =1,\;\lambda <\lambda_{\mathrm{crit}}\\ \xi =1+\alpha_{\xi ,\lambda }\left(\lambda -\lambda_{\mathrm{crit}}\right),\;\lambda \geq \lambda_{\mathrm{crit}} \end{array}$$

Nonetheless, many modelling studies have continued to use the original linear relation given by Equation (79).

In spite of evidence from the aforementioned studies that a liquid-equilibrated state elevates the electroosmotic drag coefficient, some simulations assumed *ξ* = 1 uniformly [57,108]. Quoted values of *ξ*_L_ for the liquid-equilibrated membrane vary across a wide range from 2 to 5 [105]. Zawodzinski et al. measured *ξ*_V_ = 1 and *ξ*_L_ = 2.5 at *T* = 30 °C [80], which has been subsequently applied in the Weber–Newman model implementation [65]. A polynomial fit spanning both VE and LE conditions was used by Meier and Eigenberger [58]:(81)$$\xi =1+0.028\lambda +0.0026\lambda^{2}$$

A non-linear correlation has recently been presented for Nafion 212 (a related material in the Nafion 1100 family) [95]:(82)$$\xi =1.1+\frac{0.9}{1+\exp\left(-2\left(\lambda -5.5\right)\right)}$$

The experimental study by Ge et al. gave a detailed polynomial fit for electroosmotic drag coefficient as a function of water content [109]. The data were derived from experiment using a model assuming a given explicit diffusivity (measured separately [110]) that lacked a detailed water content dependence, together with an interfacial resistance. Since any imprecision in the chosen diffusivity expression would carry forwards into the water content dependence of the electroosmotic drag coefficient, these data seem uncertain.

Data from both Ge et al. [109] (water flux measurements) and Ise et al. [111] (electrophoretic NMR) suggested a linear temperature dependence of electroosmotic drag coefficient in the fully liquid-equilibrated limit, and negligible temperature dependence otherwise (coefficient data tabulated in Table 8):(83)$$\xi_{\lambda \to \lambda_{\max}}=\xi_{0}+\alpha_{\xi ,T}\left(T-T_{0}\right)$$

These temperature-dependent expressions for electroosmotic drag coefficient are plotted in Figure 8.

Benziger et al. presented data from a hydrogen–hydrogen cell showing that even when interfacial or bulk transport limits the current density under high overpotential (e.g., due to limited hydrogen availability for reaction), water flow can continue to grow with greater applied voltage [112]. The implications of these results are not yet clear, especially as might apply to a PEMFC context rather than a hydrogen–hydrogen cell. The original work does not justify them in detail theoretically, but these observations do merit further investigation by comparison to non-equilibrium thermodynamic theories such as the Weber–Newman model.

We note in this context that the interpretation of water transport presented by Berning et al. [113] is incorrect: the authors mistakenly argue that there is no contribution from electroosmotic drag to water transport in the bulk of the membrane, on the basis that a constant electroosmotic drag coefficient allows an equation for water transport to be written that contains only the diffusional term. Even in this case, electroosmotic drag will still contribute to net water flux, and hence to the rate of sorption/desorption in the catalyst layers; thus electroosmotic drag retains a role in determining the overall water content profile and it is not correct to argue that diffusion is the only active water transport process in the bulk membrane.

Recently, Berg and Stornes [114] combined a random pore network model with the detailed swelling model of Eikerling and Berg [44] to predict a variety of apparent experimental water flux to proton flux ratios, and suggested that a ‘consistent’ thermodynamic model following Dreyer et al. [115] predicts that this experimental ratio of water flux to proton flux should tend to zero in the limit of a membrane consisting of sub-nanopores of negligible size. As far as we are aware, no continuum model has been developed to test the origin of this prediction.

### 5.7. Water Transport: Liquid-Equilibrated Conditions

Given the multiphase nature of the membrane, there has been historical disagreement over whether the definition of water content from (7) is relevant or sufficient for describing the process of diffusion down a chemical potential gradient for dissolved water. Some models account also for hydraulic flow of liquid water under a pressure gradient, in addition to diffusion down the water concentration gradient; these are discussed further below under Section 7.1, “Hydraulic Transport of Water (Flow)”. One early example is the Bernardi–Verbrugge model in which Schlögl’s equation is used to express the water flux as the sum of an electroosmotic term and a hydraulic term, with no diffusional contribution [116]; a later assessment of this work recognises that the water transport model “formulated to simulate the cathode and its gas diffusion layer” was extended to the membrane model in identical form simply “for the sake of integrity” [11], in spite of the different governing water transport phenomena in the different PEMFC regions. Weber and Newman [65] as well as Wu and Berg [104] argue that models implicating a pressure gradient become relevant only in the LE mode where free liquid water is present in the membrane.

Weber and Newman assumed that the LE conductivity becomes constant at its maximum VE value; that is:(84)$$\kappa_{L}=\kappa_{V,a_{w}=1}$$

Instead of (83), Weber and Newman suggested an Arrhenius relation for the electroosmotic drag coefficient of the liquid-equilibrated membrane [65]:(85)$$\xi_{L}=2.55\exp\left(\theta_{\mathrm{EOD}}\left(\frac{1}{T_{0}}-\frac{1}{T}\right)\right)$$
with *T*_0_ = 303.15 K and *θ*_EOD_ = 481.1 K. This relation is plotted in Figure 8.

## 6. Interface-Specific Phenomena

### 6.1. Interfacial Proton Transport Resistance

Interfacial proton transport resistance between the CL and membrane is normally considered to be negligible. Pivovar and Kim reported an experimental measurement of the ionic resistance associated with the CL–membrane contact, for a directly painted Nafion-based membrane-electrode assembly (MEA), as being at least 8 mΩ cm^2^ [117]; however, results from commercially manufactured MEAs were not reported and it has subsequently been suggested by the same authors that commercial MEAs are likely free from such resistances, which were attributed to poorly correlated swelling magnitudes under water uptake between the CL and bulk membrane [118]. Interfacial proton transport and electrical contact resistances may also increase at low water content due to a reduced membrane thickness causing mechanical decohesion of the membrane from the electrode.

The presence of an interfacial proton transport resistance even in the presence of liquid equilibration has been identified by Tsampas et al., who raise the possibility that the measured ohmic resistance of the membrane is, in fact, dominated by an interfacial membrane resistance for proton transfer across the liquid water-membrane boundary [119]. It is not clear that this study has been considered in any subsequent continuum modelling work, or whether its conclusions still apply to a membrane-impregnated CL, where other resistances may be appreciable. Evidence for significant interfacial contributions to membrane conductivity measurements has also been presented by Rangel-Cárdenas and Koper [120].

### 6.2. Interfacial Water Transport Resistance: Vapour-Equilibrated Conditions

In principle, there may be a kinetic barrier to equilibration of the membrane with adjacent phases (vapour or liquid) which would manifest itself as an interfacial resistance to water transport at the surfaces of the membrane. Interfacial transport resistances are typically measured by performing water transport measurements at a range of membrane thicknesses and then extrapolating the measured resistance to zero thickness: the intercept gives the interfacial resistance while the gradient gives intrinsic properties [121,122].

Many studies of PEMFCs have ignored interfacial transport resistance and instead assumed that the membrane surfaces equilibrate instantaneously according to the sorption isotherm [39,57,82,94,96,108,123,124,125]. This equilibrium assumption enables a numerically advantageous mathematical transformation that simplifies the overall water conservation equation by using an effective (fictitious) equivalent gas-phase concentration (as predicted from the sorption isotherm) to represent water content throughout the membrane. Within an equilibrium model, Janssen proposed that under super-saturated conditions in the membrane, both a vapour phase and a liquid phase must also be present for bulk water [96].

Also, in its original formulation, the Weber–Newman model defined the equilibrium with the vapour environment (with water activity *a*_w,vap_) as [65]:(86)$$\mu_{w}-\mu_{0,w}=RT\ln a_{w}=RT\ln a_{w,\mathrm{vap}}$$
where the second equality implies that interfacial resistance is ignored. For the case of membranes where one face is vapour-equilibrated but the other is liquid-equilibrated, interfacial effects under a humidity gradient have been argued to dominate pressure gradient effects in other works [122,126,127]. This case implicitly includes alcohol-fuelled devices, also.

Some experimental studies have suggested that relaxation of a membrane to a true equilibrium state when in equilibrium with vapour may take days to weeks, which would not be compatible with experimental timescales: the apparent equilibria maintained in PEMFC experiments might be more properly understood as quasi-equilibria, whose properties may nonetheless be determined [1]. Magnetic resonance imaging (MRI) measurements by Teranishi et al. showed a cell water content response towards equilibrium occurring on the order of 100 s after cell startup, which while much shorter than the above is still a significant duration compared to typical timescales of interest for transient PEMFC phenomena [69,128]. This is comparable to membrane and CL water uptake timescales indicated by electrochemical measurements [52,69,129] as well as to membrane sorption equilibration times [130]. Overall membrane resistance and current density were shown to respond to a potential step on the order of 1 h by Cheah et al. [121]; it is worth noting, however, that this work described an MEA based on Nafion 115, whose dimensions and total water content (by mass per unit electrode area) greatly exceed those in more modern, thinner membranes, for which equilibration following galvanostatic steps or rapid changes in inlet humidity is empirically expected to be significantly more rapid (≤1 min, see e.g., [131] with some variation reported between wetting and drying).

A general approach used to account for interfacial resistance to water transport is the inclusion of a linear mass transfer coefficient relation for the water flux at the vapour-membrane boundary:(87)$$N_{w}\cdot n=k_{\mathrm{int}}\left(a_{w}-a_{w,\mathrm{vap}}\right)$$
where **n** is the outward normal unit vector from the membrane towards the adjacent vapour phase, and *k*_int_ is an interfacial mass transfer coefficient (mol m^−2^ s^−1^). This expression was used for dehydration of the super-saturated membrane by Futerko et al., while assuming that condensation under the vapour or liquid phase maintained equilibrium [60]. In an experimental measurement at zero current, Monroe et al. measured *k*_v_ = *k*_int_(*RT/p*_sat_) ≈ 6.3 × 10^−3^ m s^−1^ at *T* = 50 °C for the VE membrane [132]. Klika has suggested *k*_int_ = 4.4 × 10^−3^ mol m^−2^ s^−1^, which seems compatible in order of magnitude.

Alternatively, some treatments have required that the water uptake derives directly from faradaic proton current density **i**_far_ across the boundary, without any chemical equilibration [101]:(88)$$N_{w}\cdot n=\frac{\alpha }{F}i_{\mathrm{far}}\cdot n$$
where *α* is a number of water molecules transferred per proton generated through the reaction, which is not necessarily equal to *ξ*.

Okada et al. considered the direct sum of an active uptake (electroosmotic) and a passive uptake (independent of current density), such that [99]:(89)$$N_{w}\cdot n=\frac{\xi }{F}i_{\mathrm{far}}\cdot n+k_{\mathrm{int}}\left(a_{w}-a_{w,\mathrm{vap}}\right)$$

The presence or absence of the active uptake term, given directly from the electroosmotic drag coefficient, has not generally been agreed upon in the literature, and merits further study. One prior study has suggested loosely but without clear justification that the electroosmotic drag term should be absent at the anode but present at the cathode [133]; for a case in which the electroosmotic drag coefficient is *excluded* from the interfacial sorption rate as specified in (89) in both CLs, cathode drying has been predicted under conditions of a low volumetric interfacial area in the CL, leading to a low effective value of *k*_int_ [134].

A similar model due to Berg et al. defined an uptake model for the VE case as linear in the difference of water contents, rather than activities [33,135]:(90)$$N_{w}\cdot n=\frac{\xi }{F}i_{\mathrm{far}}\cdot n+k_{\mathrm{int},\lambda }c_{f}\left(\lambda -\lambda_{\mathrm{eq}}\right)$$

The best experimental fit by Berg et al. gave *k*_int,*λ*_ ≈ 5 × 10^−6^ m s^−1^ [33]. Ge et al. offered measurements as a function of water content and discovered an asymmetry between absorption and desorption rates, while suggesting that *k*_int,*λ*_ ≈ 10^−5^ m s^−1^ to the closest order of magnitude and that both absorption and desorption rate can be treated as linearly dependent upon water volume fraction (determined from (9)) [110]. The temperature-dependence of *k*_int,λ_ can be expressed by an Arrhenius relation with activation energy in the range 25–31 kJ mol^−1^ [136].

Alternatively, a volumetric rate constant can be expressed as:(91)$$k_{\mathrm{vol},\lambda }=k_{\mathrm{int},\lambda }a_{\mathrm{vol}}$$
where *a*_vol_ (m^−1^) is the specific surface area of the membrane-vapour contact in the CL. Reported values include: *k*_vol,λ_ = 1.3 s^−1^ [137]; *k*_vol,λ_ = 5.7 s^−1^ [135]; *k*_vol,λ_ = 1 s^−1^ [36]; 50 s^−1^ ≤ *k*_vol,λ_ ≤ 80 s^−1^ [35]. Kosakian et al. presented fitted data at *T* = 30 °C with *k*_vol,λ_ ≈ 0.1 s^−1^ for sorption and ≈ 1 s^−1^ for desorption; this work also took the rate as proportional to the volume fraction of the vapour phase, and assumed an activated process with *E*_A,sorp_ = +20 kJ mol^−1^ [75]. Vorobev et al. performed a phenomenological study of the role of this rate constant within a CL model [138].

Klika et al. have clearly demonstrated [63] that absorption-desorption asymmetries [110,136] in fits to (90) can be explained by understanding that the true thermodynamic driving force is the activity difference as expressed in (89); the nonlinearity of the sorption isotherm then explains the resulting asymmetry when expressed in terms of water content or concentration difference. This criticism, as supported by experimental evidence, suggests that (89) should be preferred [110,132].

Kienitz et al. (originally published under the lead author name “Kientiz [*sic*] et al.”) explored the idea that interfacial resistance is itself a humidity-dependent quantity, with an investigation on Nafion 21*x* membranes, and attributed the liquid–vapour interface at a hydrophobic surface to interfacial resistances [122]. To account for this, they proposed the following expression:(92)$$k_{\mathrm{int}}{/\;\mathrm{mol}\;m}^{-2}{\;s}^{-1}=1.04\times 10^{-3}\exp^{4.48\times 10^{-4}a_{w}}$$

### 6.3. Interfacial Water Transport Resistance: Liquid-Equilibrated Conditions

Experiments suggest that LE conditions cause negligible water transport resistance at the liquid contact to the membrane [1,122,132]. The Weber–Newman model argues that there exist two membrane morphologies depending on whether or not liquid water is present [42]. In the absence of liquid water, linked clusters are connected by ‘collapsed channels’, accounting for electroosmotic drag at *ξ* = 1 due to the transport of H_3_O^+^ as the active proton carrier ion. In the presence of liquid water, channels open so that water can flow under a pressure gradient. For the intermediate regime, Weber and Newman defined *S*_L_ as a fraction of expanded channels and then used this quantity to take a linear average between the predictions of the VE and LE transport models, where the VE and LE cases have different transport properties: the consequent doubling of the number of required inputs has been identified as a limitation to the use of this model by the review of Jiao and Li [10]. The corresponding transport equations are (subscripts V and L denoting VE and LE membrane proportion, respectively):(93)$$\begin{array}{l} i=\left(1-S_{L}\right)i_{V}+S_{L}i_{L}\\ N_{w}=\left(1-S_{L}\right)N_{w,V}+S_{L}N_{w,L} \end{array}$$
where the fluxes under each case are given by (46) and (47), with transport coefficients taking different definitions for VE or LE membrane conditions.

For the liquid-equilibrated membrane, the water content is defined directly as:(94)$$\lambda =\lambda_{\max,V}+S_{L}\left(\lambda_{\max,L}-\lambda_{\max,V}\right)$$

The fraction of expanded channels is evaluated by assuming a specific pore-size distribution of hydrophobic channels in the membrane, such that the proportion of expanded channels can be expressed from the hydraulic pressure. An integration is performed across a log-normal pore-size distribution with the lower bound given by a pressure-dependent critical radius which, as developed in a later refinement of the model, has hydrophobic properties parameterised by a general energetic parameter Γ = 4 × 10^−5^ N m^−1^ [65]:(95)$$S_{L}=\int_{r_{\mathrm{crit}}}^{\infty }V(r)dr=\frac{1}{2}\left(1-\mathrm{erf}\left(\frac{\ln\left(r_{\mathrm{crit}}/\mathrm{nm}\right)-\ln1.25}{0.3\sqrt{2}}\right)\right)$$
(96)$$r_{\mathrm{crit}}=\frac{\Gamma }{p_{L}}$$

In the critical radius Equation (96), the liquid water pressure at the membrane-vapour interface (*p*_L_) is then determined by mass conservation: the specific form will depend also upon the GDL model, which was developed by later authors after some simplistic assumptions in early works [139]. The work by Weber and Newman is somewhat vague about how the degree of saturation is defined when a membrane is liquid-equilibrated only in one spatial region: any implementation of the above approach in such a case seems to require additional assumptions [36,140].

Meng et al. implemented the liquid saturation by means of an adapted Springer model [141]:(97)$$\lambda_{\mathrm{eq}}=14+2.8S_{L},\;S_{L}>0$$
noting that the saturation extent here references an equilibrium water content rather than an actual water content. This work argues that once liquid water channels are opened in the membrane, they can accommodate additional free liquid water which is not bonded, while the membrane accommodates a given quantity of water per (97)—puzzlingly, this approach seems to count the additional water present due to liquid-equilibration twice over, because of the presence of the saturation in (97). In turn the liquid saturation evolves according to a capillary diffusion equation:(98)$$\nabla \cdot \left(D_{\mathrm{cap}}\nabla S_{L}\right)-\nabla \cdot \left(\frac{S_{L}^{3}\kappa_{p}}{v_{w}}\nabla p\right)=R_{w}$$
where *D*_cap_ is capillary diffusivity (set as a constant = 2 × 10^−5^ kg m^−1^ s^−1^) and *κ*_p_ is a permeability for the membrane (set = 1.8 × 10^−18^ m^2^ following Bernardi and Verbrugge [142]). The water source *R*_w_ is expressed in terms of the balance of condensation of vapour to liquid water and bonding of water to the membrane according to the sorption equilibrium (97):(99)$$R_{w}=k_{\mathrm{vap}}M_{w}\left(p_{\mathrm{vap}}-p_{\mathrm{sat}}\right)-k_{\mathrm{bond}}M_{w}c_{f}\left(\lambda_{\mathrm{eq}}-\lambda \right)$$
with *k*_bond_ = 1 s^−1^ [36]. The value for *k*_vap_ is not clearly specified in the original work but for a typical specific surface area of vapour–liquid contact, the work of Wu et al. implies a value of the order of 7000 s m^−2^ kg^−1^ [36].

Hwang et al. employed the van Genuchten model for the degree of saturation in partially saturated porous media to describe the membrane as governed by capillary hydrodynamics (detailed formulas are given in the referenced work) [143]. This assumed a van Genuchten parameter *n* = 3.56 for Nafion, without any real justification. At the boundary between porous media properties, this work asserted continuity of liquid and vapour pressure, and hence discontinuity of saturation. Due to the lack of morphological similarity between Nafion and the porous rocks on which the van Genuchten model is based, this approach seems doubtful, and it does not appear to have been followed subsequently.

The works of Wu et al. and Falcão et al. distinguished between the anode as being exclusively vapour-equilibrated, and the cathode side, where the water produced through the cathode reaction (oxygen reduction) is initially membrane-dissolved, as liquid-equilibrated [36,144]. The frequent observation of net cathode-to-anode water transport in the presence of a water-saturated anode fuel has been described as a manifestation of Schröder’s paradox due to the liquid equilibration of the cathode [127]. Adachi et al. demonstrated experimentally that allowing liquid equilibration of one face of the membrane, while maintaining vapour equilibration on the other, greatly increased the permeation rate compared to a vapour–vapour membrane, due to the elimination of one interfacial resistance [126].

## 7. Mechanical Phenomena

### 7.1. Hydraulic Transport of Water (Flow)

Hydraulic transport of water is dependent upon a continuous liquid water phase within the membrane, which according to the physical model of Weber and Newman only arises under LE conditions [42]. Models that focus on conditions of vapour equilibration, especially those based on the Springer model and its transient extension, may ignore pressure gradients altogether [39,124,145]. The Weber–Newman theory argues with support from Janssen that liquid water content, as distinct from water content dissolved in the membrane, must be modelled explicitly to allow a pressure gradient to drive liquid water flow alongside the chemical potential gradient [9,96]; this point is analogous to the analysis by Kreuer et al. which showed that the liquid water chemical potential gradient must vanish in the presence of connected channels containing bulk liquid [2]. A significant increase in water transport rate under LE conditions has been demonstrated experimentally [130].

The quasi-empirical Schlögl equation used for water transport in the works of Bernardi and Verbrugge assumes the presence of liquid water filling the membrane pores, and sets [116,142]:(100)$$\frac{\mu M_{w}}{\rho_{w}}N_{w}=-\kappa_{p}\nabla p-\kappa_{\phi }c_{f}F\nabla \phi$$
where *μ* is viscosity, *κ*_p_ and *κ*_φ_ are the hydraulic and electroosmotic permeabilities respectively, and *p* is pressure. This equation is unsuitable in the VE regime where the assumption of the presence of liquid water is not valid [65,96]. The original data (at *T* = 80 °C) are: *μ* = 3.56 × 10^−4^ kg m^−1^ s^−1^; *κ*_p_ = 1.8 × 10^−18^ m^2^; (*κ*_φ_
*c*_f_) *=* 8.616 × 10^−17^ mol m^−1^. Hydraulic permeability was given as 5 × 10^−19^ m^2^ by Nam et al. [146]. Meier and Eigenberger used experimental data to measure the following hydraulic permeability at 25 °C:(101)$$\kappa_{p}/m^{2}=10^{-20}\left(0.38+0.04\lambda +0.014\lambda^{2}\right)$$

The works of the Djilali group have criticised features of the “dusty fluid” model combining hydraulic and diffusive transport, as developed by Thampan et al. [30], as being physically unreasonable for the PEMFC context [77,82]. In particular, this model is criticised for double-accounting flux contributions by imposing Schlögl’s equation for viscous phenomena convective velocity on top of a Maxwell–Stefan equation, which already considers all contributions to the velocity of each component species [78]. The similar non-equilibrium thermodynamic model presented by Rama et al. gave independent diffusion and hydraulic pressure gradient terms in the water flux expression [147]; this would contradict the same principles of fundamental multi-component transport theory, and this model has not been taken up by other researchers. Eikerling et al. [97] argued for the predominance of convective transport according to (100), but emphasised that sorption equilibrium meant that the pressure could be expressed directly in terms of water content, and hence the pressure gradient term is not distinguishable from an effective diffusivity expressed in terms of water content, as in the Springer model.

In the Weber–Newman model, the liquid-equilibrated gradient of water chemical potential in (47) is expressed as [65]:(102)$$\nabla \mu_{w,L}={\bar{V}}_{w}\nabla p$$

Hence, the LE water flux becomes a hydraulic flow [65]:(103)$$N_{w,L}=-\alpha_{w,L}\nabla p+\frac{\xi_{L}}{F}i$$

The mass transfer coefficient is assumed to take the value:(104)$$\alpha_{w,L}=\frac{\kappa_{p}}{\mu {\bar{V}}_{w}^{2}}{\left(\frac{\phi_{w}}{\phi_{w,\max}}\right)}^{2}$$

In the above, the viscosity *μ* should be understood to be its bulk value for liquid water, which is a general function of temperature as reported in standard engineering data sources.

### 7.2. Membrane Expansion and Mechanical Constraint

Membrane materials are known to undergo swelling as their water content increases. In the unconstrained material, the inelastic volumetric expansion can be expressed using (12). In a PEMFC, however, the membrane is constrained through compression, and so is not free to expand; consequently, membrane strain in a PEMFC depends on the overall mechanical properties of the device, and the extent of hydration. It has also been reported that membrane expansion is likely to compress the GDL of the operating PEMFC due to the higher stiffness of the membrane compared to the GDL [13,148]. Investigation of the complex structural interactions between laminated MEA components has indicated that stress–strain measurements on free membranes are unlikely to be representative of in situ behaviour [149].

Experimental measurements of the inelastic expansion strain due to membrane swelling suggest a magnitude of the order of 0.01 per unit *λ* in unreinforced membranes [1,39,150]; corresponding measurements on reinforced expanded polytetrafluoroethylene (ePTFE) GORE-SELECT membranes gave inelastic expansion strains about 5 times lower [151]. Both experimental and theoretical studies have implicated the load due to membrane swelling in a lowering of the effective water content of the membrane [128,148,152], with an experimental cap on water uptake of *λ*_max_ = 6.5 reported in one case [128].

The formulation of mechanical constitutive relations (e.g., hyperelastic and/or viscoelastic-plastic constitutive models) for hydrated Nafion materials has been a subject of several works but, since such models are seldom if ever combined with practical electrochemical device models, a detailed discussion falls outside the scope of this review; the interested reader is directed to some key publications for further information [150,153,154,155]. It has also been established that the CL and interfacial properties may have a significant role in overall structural behaviour of the membrane [156].

To assess the influence of mechanical stress on water transport in an operating PEMFC, Weber and Newman proposed that a membrane within a PEMFC MEA can be described by a degree of constraint *χ*_c_, such that the membrane volume change under hydration is zero with *χ*_c_ = 1 and equals its free, unconstrained value with *χ*_c_ = 0 [148]:(105)$${\bar{V}}_{\mathrm{mem}}={\bar{V}}_{p}+\lambda {\bar{V}}_{w}\left(1-\chi_{c}\right)$$

Considering that the constraint introduces an associated stress that will contribute to the chemical potential of the water, this work assumed a balance between the chemical potential for water inside and outside the membrane in terms of membrane bulk modulus, and yielded a comparison to the unconstrained case as [148]:(106)$$\frac{\lambda_{\mathrm{constrained}}}{\lambda_{\mathrm{free}}}={\left(\frac{{\bar{V}}_{\mathrm{mem}}}{{\bar{V}}_{p}+\lambda {\bar{V}}_{w}}\right)}^{\frac{Y_{\mathrm{mem}}{\bar{V}}_{w}}{3RT}}$$
where *Y*_mem_ is the Young’s modulus of Nafion 1100, expressed as:(107)$$Y_{\mathrm{mem}}=Y_{0,\mathrm{mem}}\frac{T_{0}}{T}\exp\left(-0.1655\left(12-10\left(M_{\mathrm{EW}}/\mathrm{kg}{\mathrm{mol}}^{-1}\right)+\frac{\lambda M_{w}}{M_{\mathrm{EW}}}\right)\right)$$
where *Y*_0,mem_ = 275 MPa and *T*_0_ = 303.15 K. This work then used an iterative simulation procedure, beginning from the unconstrained case, in order to predict the self-consistent value of the constrained water content from (106).

Kusoglu et al. used a Flory–Huggins model for the sorption thermodynamics together with the Mori-Tanaka model for bulk modulus of a two-phase material, in order to provide an overall sorption isotherm under pressure, with experimental corroboration [152]. Both Kusoglu et al. and Klika et al. have argued that the influence of pressure on contact resistances will have a greater practical impact than variation in uptake or transport due to swelling phenomena [63,152]. Hasan et al. incorporated in isotropic swelling model combined with a viscoelastic-plastic mechanical model into an electrochemical analysis: this allowed strain to be predicted as a function of membrane hydration, but no feedback from the mechanical response was considered to the water transport and uptake [31].

## 8. Transient Response of the Membrane

As noted in Section 6.2 above, sorption and water transport phenomena may not equilibrate rapidly compared to relevant experimental timescales. Additionally, intrinsically dynamic experimental methods such as electrochemical impedance spectroscopy (EIS) are of interest for PEMFC characterisation, and require corresponding simulation development for interpretation of their results [75,88,95,157]. For this reason, some authors have explored transient extensions to the quasistatic membrane models discussed thus far.

In general, interfacial resistances and dynamics of sorption may dominate the membrane response to a perturbation in its surroundings [1]. Therefore, diffusion coefficients reported from mass-uptake experimental methods must be treated with great care [130]. The consistent incorporation of interfacial phenomena and liquid equilibration effects described in Section 6.2 is the best strategy to allow for a correct transient prediction.

The first significant development of a transient continuum membrane model was the extension of the model established by Um et al. [123] to a transient model [124]. This defined a transient water content balance equation for the membrane phase as:(108)$$c_{f}\frac{\partial \lambda }{\partial t}+\nabla \cdot N_{w}=0$$

This equation applies for the pure membrane region where no electrochemical reaction takes place; in the CL, a source term could apply due to the rate of sorption/desorption. Transient volume changes associated with swelling are not considered self-consistently in the form (108).

In general it is common to consider that the relaxation timescales for proton conductivity are significantly more rapid than other transport processes, and so a quasistatic proton conduction model can be used in conjunction with transient transport models for water content and for species transport in other regions of the PEMFC [158]. Thus, the stationary proton conduction Equation (6) is applied alongside (108) in the typical transient formulation of the Springer model [124,159]. Ziegler et al. extended the Weber–Newman model to consider transient systems [160], using the explicit Thampan isotherm (Equation (30)).

## 9. Non-Isothermal Phenomena

Because of the thin spatial dimension of the membrane, it is standard to assume that heat transfer is dominated by thermal conduction, and maintains a quasi-steady state. The following equations then apply to describe the heat flux **q**:(109)∇⋅q=Q
(110)q=−k∇T

Thermal conductivities (*k*) for Nafion 1100 have been measured variously in the range 0.1–1 W m^−1^ K^−1^; data have been reported as a function of temperature and water content [12,161]. The net heat source *Q* arises within the membrane due to resistive heating [84]; for an Ohm’s law treatment (Equation (5), as used in the Springer model), the corresponding Joule heat source is:(111)$$Q=\frac{i^{2}}{\kappa }$$

Temperature dependence of the transport coefficients was discussed above in Section 5. The proton conductivity and water diffusivity are generally agreed to obey Arrhenius equations in the range 50 °C < *T* < 80 °C [1,59,65].

For a more general non-isothermal case, it is necessary to consider thermoosmosis—that is, the transport of water under a temperature gradient. Dai et al. (writing in 2009) suggested that thermoosmotic transport of water in the membrane was not well understood [162], but could contribute appreciably due to internal temperature gradients, even in the presence of good thermal balance between bipolar plates.

A standard equation expresses this additional contribution in terms of a thermoosmotic diffusion coefficient *D*_w,*T*_:(112)$$N_{w}=N_{w,\mathrm{const}T}-D_{w,T}\nabla T$$

Even the direction of this effect is uncertain: the sign of *D*_w,*T*_ should depend on the relative hydrophilicity or hydrophobicity of the membrane. Some experimental studies have suggested *D*_w,*T*_ < 0 [163] while others have suggested *D*_w,*T*_ > 0 [164]. The former measurements also gave an Arrhenius behaviour for *D*_w,*T*_ with an activation energy comparable to that for mass diffusion of water in Nafion 1100 (as Equation (72)); this was incorporated into a recent full cell study as [75]:(113)$$D_{w,T}=D_{w,T,0}\exp\left(-\frac{\theta_{\mathrm{diff},T}}{T}\right)$$
with *D*_w,*T*,0_ = −1.04 × 10^−5^ kg m^−1^ s^−1^ K^−1^ and *θ*_diff,*T*_ = 2362 K.

Within the same experimental studies, it has been argued that thermoosmosis is sufficiently negligible to make it reasonable to approximate that *D*_w,*T*_ = 0 [163,164]. In this theory, the perceived contribution to mass flux due to temperature gradients in fact arises due to other mechanisms: (a) ‘heat piping’ due to the difference in water saturation pressure on the two faces of the membrane driving condensation and evaporation at cold and hot faces, respectively; (b) temperature dependence of the sorption isotherm, introducing a diffusive driving force at constant water content. To account for the latter, Fu et al. introduced the following modification to (97) [164]:(114)$$\lambda_{\mathrm{eq}}=9.2+\left(0.18+0.138\left(T-T_{0}\right)\right)S_{L},\;0<S_{L}\leq 1$$

## 10. Transport of Other Chemical Species

### 10.1. Dilute Gas Transport

In device models, it is most common to treat the membrane as strictly gas-impermeable. As membranes have become thinner, however, gas crossover has become an increasingly important phenomenon to understand quantitatively. For PEMWEs, gas crossover effectively limits the extent to which the membrane can be thinned; for PEMFCs, it is important when considering the gas composition in the anode recirculation loop, and for investigating radical formation.

Wherever an explicit description of gas crossover is required, the flux of dilute dissolved gas (**N**_gas_) across the membrane can be expressed empirically using permeation coefficients (*ψ*_gas_) and the difference in partial pressure on the two faces of the membrane, as related to the membrane thickness:(115)$$N_{\mathrm{gas}}=-\psi_{\mathrm{gas}}\nabla p_{\mathrm{gas}}$$

In such gas crossover models, it is common to assume an infinitely rapid reaction of H_2_ at the cathode and O_2_ at the anode, such that the concentrations of the dissolved gases go to zero at the respective boundaries [146].

Values of *ψ*_gas_ are typically of the order 10^−15^ to 10^−14^ mol m^−1^ s^−1^ Pa^−1^, with values in pure water closer to 10^−13^ mol m^−1^ s^−1^ Pa^−1^ [1]. For the common PEMFC gases, *ψ*_H2_ > *ψ*_O2_ > *ψ*_N2_. Weber suggested the following approximate relative relations [165]:(116)$$\begin{array}{c} \psi_{O_{2}}=\psi_{N_{2}}=\frac{2}{3}\psi_{H_{2}}\\ \psi_{H_{2}O}=\frac{8}{9}\psi_{H_{2}} \end{array}$$

This work argued that the threshold for significant performance impact arises at 10^−13^ mol m^−1^ s^−1^ Pa^−1^, which is appreciably higher than the measured permeation coefficients in wet Nafion [165]. Kundu et al. measured *ψ*_H2_ ≈ 7.4 × 10^−14^ mol m^−1^ s^−1^ Pa^−1^ in a GORE PRIMEA series 5510 MEA [166].

Zhang et al. measured *ψ*_O2_ ≈ 1.6 × 10^−14^ mol m^−1^ s^−1^ Pa^−1^ through Nafion 117 at *T* = 80 °C and gave an activation energy for permeation of 23 kJ mol^−1^ in the range 40 °C < *T* < 100 °C [167]. Earlier measurements had suggested a comparable activation energy for O_2_ permeation of the order 30 kJ mol^−1^ [168]. Various studies have reported Nafion 1100 permeation data additionally for CO_2_ and N_2_, including the functional relationship with relative humidity and temperature [169,170]. Studies have indicated particularly low permeability for N_2_ (<10^−15^ mol m^−1^ s^−1^ Pa^−1^) both through Nafion 1100 and GORE PRIMEA catalyst-coated membranes [170,171].

Weber and Newman compiled then-available experimental data to give the following expressions for H_2_ and O_2_ permeability under VE and LE conditions, with water content-dependence in the former case (coefficient data are tabulated in Table 9, and permeation coefficients are plotted against temperature in Figure 9) [65]:(117)$$\psi_{i,V}=\left(\psi_{w,i}\phi_{w}+\psi_{V0,i}\right)\exp\left(\frac{E_{A,DV,i}}{R}\left(\frac{1}{T_{0}}-\frac{1}{T}\right)\right)$$
(118)$$\psi_{i,L}=\psi_{L0,i}\exp\left(\frac{E_{A,DL,i}}{R}\left(\frac{1}{T_{0}}-\frac{1}{T}\right)\right)$$

The permeation coefficient is a convenient measure due to its combination of the solubility of the gas and the diffusivity of dissolved gas in the membrane into a single empirical quantity [9]. Since these effects often have opposing temperature dependences, with solubility falling with temperature while diffusivity rises, permeability is only weakly dependent on temperature, albeit still positively increasing.

Where a diffusion model is required for the dissolved gas, it is common to assume that the dissolved gas concentration (*c*_gas_) is low enough that Fick’s law can be applied [9]:(119)$$N_{\mathrm{gas}}=-D_{\mathrm{gas}}\nabla c_{\mathrm{gas}}$$

Here the diffusion coefficient *D*_gas_ relates to the permeation coefficient as:(120)$$D_{\mathrm{gas}}=\frac{\psi_{\mathrm{gas}}}{K_{H,\mathrm{gas}}}$$
where *K*_H,gas_ is the Henry’s law coefficient such that, at equilibrium:(121)$$c_{\mathrm{soln}}=\frac{p_{\mathrm{gas}}}{K_{H,\mathrm{gas}}}$$

Wong and Kjeang compiled solubilities from a variety of prior sources as follows (coefficient data are tabulated in Table 10 and Henry’s law coefficients are plotted against temperature in Figure 10) [137]:(122)$$K_{H,\mathrm{gas}}=K_{H,\mathrm{gas},0}\exp\left(-\frac{\theta_{\mathrm{soln},\mathrm{gas}}}{T}\right)$$

Bernardi and Verbrugge expressed the gas diffusion coefficients directly, in the following temperature-dependent form (coefficient data are tabulated in Table 11) [142]:(123)$$D_{\mathrm{gas}}=D_{\mathrm{gas},0}\exp\left(\frac{-\theta_{\mathrm{diff},\mathrm{gas}}}{T}\right)$$

Within a degradation model, Wong et al. gave *D*_H2O2_ = *D*_HF_ = 1.5 × 10^−10^ m^2^ s^−1^ (at an unspecified temperature) [137]. It has been suggested that N_2_ can be treated with identical solubility and diffusion coefficient properties as O_2_ [172]. In a study of carbon corrosion, Hu et al. used *D*_O2_ = 10^−9^ m^2^ s^−1^, which is much higher than values proposed elsewhere [173].

Rangel-Cárdenas and Koper included H_2_ permeation of the membrane in a non-equilibrium thermodynamic model, but without any further application of the equations derived thereby [120].

### 10.2. Transport of Other Ions

The presence of mobile ions other than protons in the membrane complicates the description of transport phenomena discussed above. Contaminating ions may arise from membrane manufacture, degradation of PEMFC components during operation or reactant impurities. Since if multiple ionic species are present in the membrane, proton transport and current density are no longer equivalent, relation (4) does not hold; furthermore, the addition of other ionic components will mean that binary quantities such as the conductivity and diffusivity values reported above are no longer valid [174].

The simplest approach is to assume that contaminating ionic species are present in very dilute concentration, such that the Nernst–Planck equations can be used for their transport while considering the current density and apparent membrane potential to be dominated by proton transport, so that (5) or (46) still holds. Then, the flux of a dilute ionic species *i* with charge number *z_i_* is:(124)$$N_{i}=-D_{i}\nabla c_{i}-\frac{z_{i}F}{RT}D_{i}c_{i}\nabla \phi$$

Weber and Delacourt extended a concentrated electrolyte solution theory to consider the presence of a single contaminating cation [175]. Burlatsky et al. described Pt^2+^ transport with *D*_Pt_ ≈ 10^−10^ m^2^ s^−1^ at unspecified temperature [176]. Fe^2+^ and Fe^3+^ ions as contaminants have been described with diffusivities *D*_Fe2+_ ≈ 4 × 10^−10^ m^2^ s^−1^ and *D*_Fe3+_ ≈ 4 × 10^−11^ m^2^ s^−1^ at *T* = 95 °C [177,178]. The presence of metallic cations in the membrane can accelerate membrane degradation [16] both through the formation of radical species and through the precipitation of solid bands of platinum inside the membrane; chemical degradation models are discussed further below (Section 11.2), but more detailed discussion of the specific contaminated membrane case in the work of Burlatsky et al. [176] exceeds the scope of this review.

## 11. Membrane Degradation

For nearly all applications, the durability of PEMFCs is critical. For instance the US Department of Energy has fuel cell targets in the automotive sector requiring no more than a 10% loss in rated power over operating times of 8000 h for passenger cars and >25,000 h for heavy-duty vehicles [179]. Performance may be especially impacted by membrane degradation, particularly through reduced proton conductivity and through enhanced hydrogen crossover due to loss of membrane thickness or formation of pinholes. These degradation pathways may also cause other components to degrade more rapidly.

Due to the central role of the membrane in cell degradation, predictive modelling can support industrial development by facilitating prediction of the rate of degradation, and/or correlating metrics of degradation to cell performance in a more fundamental, physical manner. The relevant mechanisms of degradation are complex and varied; a detailed discussion is beyond the scope of this review. The most important phenomena can broadly be classed as either chemical aging of the membrane by radicals, or mechanical aging by repeated dimensional change of the membrane [16].

The simplest approach to incorporating general membrane degradation in a practical device model is to use a purely empirical specification of degradation rate, without correlation to other physical features of the model. Without correlation to a specifically mechanical or chemical origin of degradation, Karpenko-Jereb et al. followed this approach by specifying a constant proportional decrease in proton conductivity (and, concurrently, *c*_f_) at 5.64 × 10^−4^ h^−1^, and a constant proportional decrease in membrane thickness at 3.71 × 10^−4^ h^−1^ [172]. Once the membrane thickness reaches a critical thickness (defined as 10% of its initial thickness by the authors), pinholes are simulated by increasing gas crossover rates by an arbitrary but empirically motivated multiple of 100.

### 11.1. Mechanical Degradation Models

Mechanical degradation of the membrane may consist of pinhole and microcrack formation, membrane creep and even delamination of the CL from the membrane surface [180]. In general, models to date focus either on empirical description of the impact of degradation, or prediction of degradation rate; there are no coupled models for ongoing mechanical degradation prediction alongside performance prediction.

Weber described a membrane pinhole with radius *r*_hole_ according to a localised volume fraction *ε*_hole_ within a degraded area of the membrane *A*_deg_ [165]:(125)$$\varepsilon_{\mathrm{hole}}=\frac{\pi r_{\mathrm{hole}}^{2}}{A_{\deg}}$$

Within this region, the membrane effective properties are scaled by (1-*ε*_hole_) while the Maxwell–Stefan diffusion equations from the adjacent GDLs are extended through the membrane with porosity *ε*_hole_ and unit tortuosity (assuming a straight, cylindrical pinhole). Based on this theory, the study explored the impacts of pinholes with different sizes and frequencies along a channel length, revealing especially significant performance degradation in the case of single pinholes occupying a volume fraction *ε*_hole_ > 0.002. In a similar approach, membrane-electrode delamination over a prescribed area has also been described empirically, simply by applying an infinite contact resistance at the membrane-CL interface [181].

Burlatsky and co-workers developed a cyclic stress model in order to predict damage accrual due to fatigue from repeated hydration and dehydration of the membrane [182,183]. However, they did not provide any model of performance deterioration or the impact upon chemical properties of this degradation route. In general, membrane fatigue models have been used for durability analysis [184] but have not yet been applied to a direct prediction of performance deterioration. As mechanical stress on the membrane is closely coupled to dimensional changes, membranes with reinforcements are expected to show very different mechanical aging behaviour, and care should be taken when parameterising models using historical data.

### 11.2. Chemical Degradation Models

The dominant chemical mechanism considered in chemical degradation models of Nafion membranes is via the generation of H_2_O_2_ through the crossover reactions of dissolved H_2_ at the cathode and/or dissolved O_2_ at the anode. In the presence of Fe^2+^, which is generally present as a dispersed membrane contaminant, the Fenton reaction generates ^•^OH and ^•^OOH radicals, which lead to membrane degradation by side-chain and main-chain scission of the perfluorosulfonate membrane material. The F^−^ byproduct of the degradation reaction is a common experimental tracer of degradation rate and as such can be applied to model validation [166].

Some semi-empirical models have assumed that the rate of ^•^OH radical release is in direct proportion to the crossover flux of the contributing species (H_2_ or O_2_), and membrane degradation rate is in turn proportional to ^•^OH radical generation rate. Hence, Kundu et al. gave the variation in membrane thickness *L*_mem_ as [166]:(126)$$\frac{dL_{\mathrm{mem}}}{dt}=-k_{\deg}\psi_{H_{2}}p_{H_{2},\mathrm{ano}}$$
with *k*_deg_ = 1.8 × 10^−8^ m^2^ mol^−1^ for a GORE PRIMEA series 5510 MEA.

Chandesris et al. gave a similar semi-empirical expression, including potential and temperature dependence of *k*_deg_ and assuming thickness-dependent degradation (assuming constant reaction rate with crossover flux) [185]:(127)$$\frac{L_{\mathrm{mem}}}{L_{\mathrm{mem},0}}\frac{dL_{\mathrm{mem}}}{dt}=-k_{\deg}\left(U_{\mathrm{ano}},T\right)\psi_{O_{2}}p_{O_{2},\mathrm{cat}}$$

Pinhole formation is predicted within this model as thinner regions of the membrane degrade more rapidly, amplifying any initially present non-uniformities.

These models were extended by Shah et al. to incorporate the full chemical mechanism for the Fenton reactions and consequent ‘unzipping’ of the membrane structure [186]. In this model, independent concentrations are assigned to several constituent repeated moieties within the polymer structure (carboxylic acid, weak polymer end groups, side chains, CF_2_); as these concentrations evolve under the degradation model, they allow a prediction of spatial and temporal evolution of degradation sites. This model did not consider the consequences of degradation on relevant performance properties of the membrane, however. Subsequently, Futter et al. incorporated the role of Fe^2+^ and Fe^3+^ ion transport explicitly [178]. Burlatsky and co-workers have also considered the role of Pt reprecipitation on membrane degradation, since reprecipitated Pt from Pt^2+^ transport can agglomerate within the membrane and act as a catalytic site for peroxide-driven chemical degradation; these membrane-scale Pt reprecipitation models have not yet been combined with a full electrochemical model for the PEMFC [176,187]. We recommend that these papers be consulted directly for the detailed specification and parameterisation of the reaction mechanism.

More recent works have correlated the extent of chemical degradation predicted from a mechanistically detailed Fenton reaction model to increased gas crossover and performance deterioration. The principal approach is the definition of an ‘effective porosity’ for the membrane which varies with the concentrations of the membrane chemical moieties [188,189]. By considering *c*_f_ to be a variable depending on the sulfonate and total polymer concentrations, Wong and Kjeang defined an altered ${\bar{V}}_{p}$ for the degraded membrane, and used this to update the membrane porosity in the BFM description (using (9) and (59)) [137]. This model also altered membrane thickness directly in proportion to total mass loss, and has recently been incorporated into a 3D full cell performance model [190]. Similarly, Quiroga et al. gave a specific expression for degraded membrane molar volume according to the side-chain concentration [191]:(128)$${\bar{V}}_{p}=\frac{1}{c_{\mathrm{side}-\mathrm{chain}}{\left(1+\beta_{w}\lambda \right)}^{3}}$$

The membrane conductivity was in turn correlated to ${\bar{V}}_{p}$ according to an effective medium model; the latter was then based on a coarse-grained molecular dynamics database.

In recent years there have been advances in membranes that reduce their susceptibility to chemical degradation. These are not always explicitly indicated in theoretical descriptions of membranes and, therefore, great care should be used when parameterising degradation models using literature data from older materials. Post-fluorination of PFSA end groups and the introduction of radical scavengers into the membrane have been particularly effective, as has the use of mechanical reinforcement to reduce chemical-mechanical aging mechanisms [16].

## 12. Perspective

The ability to simulate fuel cells has been key to their recent advancement. We expect device simulation to play an increasingly important role in the rational design of new fuel cell technologies and in understanding the behaviour of these complex devices. As PEMFC technology matures, ever greater optimisation will be required to achieve meaningful performance improvements, and higher fidelity models will, therefore, be required to continue to design better fuel cells. Modelling will also play a key role in extending the operating life of PEMFCs. With widespread fuel cell deployment needed urgently to contribute to climate change targets, it is impossible to experimentally test new prototype PEMFCs for all operating conditions or end-use applications when the required lifetimes are in excess of 30,000 h (>3 years). Simulation will, therefore, be key in providing designers and users with confidence that new fuel cell designs will have the high longevity needed for use in heavy-duty applications such as trucks, trains, ships and aeroplanes. It will also be crucial in the development of the diagnostic and prognostic measurements that will be needed for the operation of PEMFCs and for the application of new measurement-integrated digitisation technologies such as ‘digital twins’.

On consulting the literature, it is clear that even models developed in the last 5 years depend extensively on heritage parameterisation from the 1990s, in spite of significant developments in industrially relevant PEMFC membrane materials since this time. While many fundamental modelling concepts continue to apply to contemporary PFSA-based membranes, parameterisation based on industrially outdated materials like Nafion 117 is increasingly irrelevant. As an instructive example, we investigated, randomly and without any *a priori* pre-selection, 8 papers published since 2019 that contain 2D or 3D PEMFC device simulations, all from different research groups [75,178,192,193,194,195,196,197]. Of these, 6 use the Springer model and 2 use the Weber–Newman model. In works using the Springer model, parameterisation of the membrane properties is exclusively drawn from four experimental works [39,52,53,87], all dating to 1998 and earlier, and all measured on Nafion 117. Data correction in CLs was limited to scaling according to an effective tortuosity. In works using the Weber–Newman model, a recent measurement was used for the proton conductivity in one case, but otherwise the data (where reported) drew from the same 1990s experimental measurements, with some rescaling for fitting purposes in one case.

On this basis, we note a general absence of models and/or data in the recent literature that explicitly address the following features of state-of-the-art PEMFC membranes:The composite nature of reinforced membranes, including its effect on conductivity, diffusivity, water uptake, and mechanical coupling to transport phenomena.The introduction of radical scavengers or other membrane ‘additives’.Parameterisation at relevant operating conditions (>60 °C), as opposed to at ambient conditions.The impact of variations in different membrane chemistries/side-chain length.The use of very thin membranes (<50 μm).In-plane inhomogeneities in electrode profiles.

We highlight these industry-relevant, contemporary membrane features as priorities for further theoretical study.

## 13. Conclusions

This review has summarised 30 years of development in the macroscopic theory of transport phenomena of polymer electrolyte membranes, as applied to practical models of polymer electrolyte membrane fuel cells. A spatial model of proton current from Ohm’s law is achievable under the assumption of uniform membrane hydration. However, the water content dependence of conductivity, as well as the intrinsic role of membrane water transport phenomena (electroosmotic drag and water diffusion) in cell water balance, encourage the vast majority of spatially-resolved PEMFC simulations to incorporate a coupled proton-water transport model in the membrane. The semi-empirical Springer and Weber–Newman models have been the most popular of these, and allow relatively straightforward extension to transient and/or non-isothermal conditions, as well as to account for interfacial resistance to water uptake or loss in the CLs. During the last decade in particular, an increasing number of predictive models have also been developed to consider the commercially relevant concerns of gas crossover, impurity transport and membrane degradation.

Accurate physical models of the charge, mass and heat transport phenomena of the membrane are essential to high-fidelity prediction of PEMFC performance and localised behaviour. In collating models with their parameterisation data in this review, and by appraising collectively the heritage of work back to the early 1990s rather than focusing only on the last few years of progress, we aim to instigate greater efforts to compare existing models to new models. In particular, such comparisons may aid in demarcating the range of conditions in which particular models are most suitable, and the tasks to which they can be applied.

The theoretical description of PFSA membranes is challenging in terms of its physics, but also represents a moving target for PEMFC modelling and its underlying experimental parameterisation. Membrane chemistries and structures will continue to advance and, considering the increasing lifetime and performance demands for PEMFCs, more accurate modelling of ‘edge-case’ transport and degradation mechanisms, such as poisoning by ions leached from bipolar plates, is likely to become ever more important. This review, therefore, highlights the need for higher-fidelity models and the high-quality fundamental experimental data on state-of-the-art materials needed to parameterise them.

Our outlook in the Perspective section underlined a present disconnection between the experimental state-of-the art and the parameterisation most often employed in practical PEMFC simulation. Thus, the task of developing PEMFC membrane models is far from complete. We strongly encourage the community to continue to develop useful, practical models of these vitally important and fascinating materials.

## Figures and Tables

**Figure 1 membranes-10-00310-f001:**
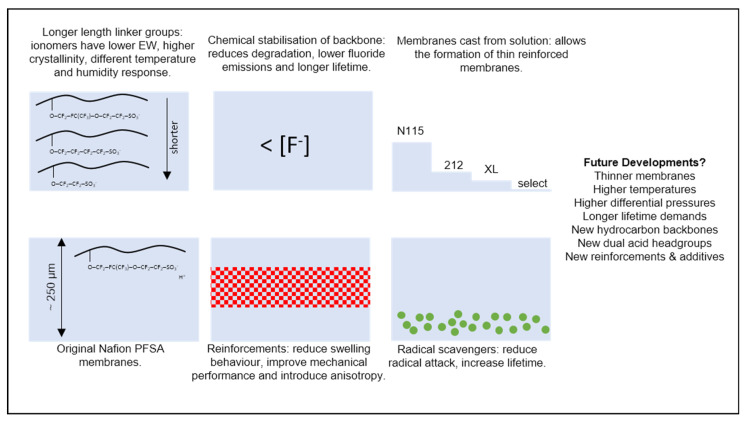
Schematic highlighting the key innovations in polymer electrolyte membrane fuel cell (PEMFC) membranes.

**Figure 2 membranes-10-00310-f002:**
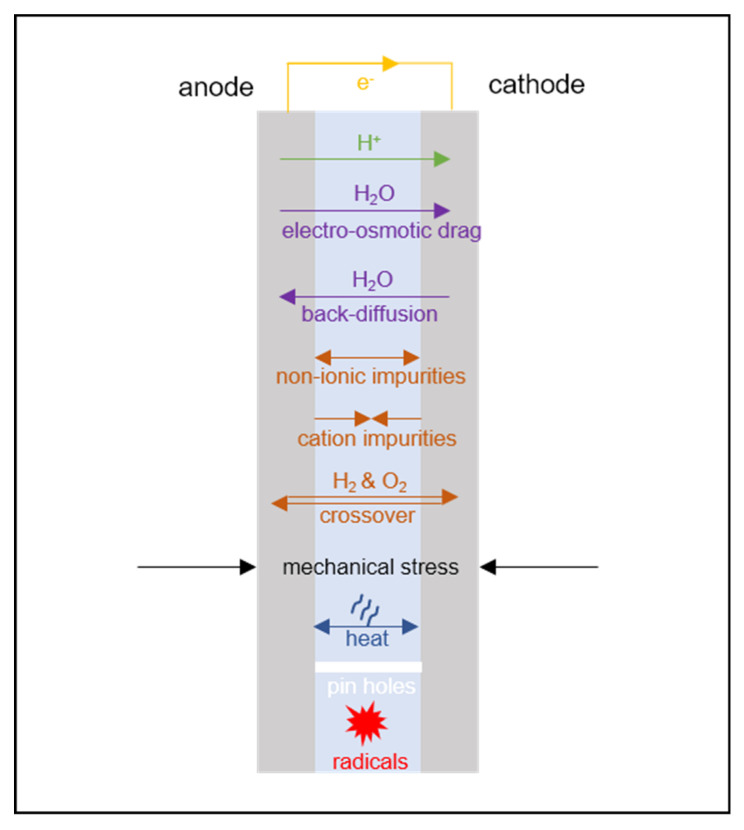
Schematic of different transport phenomena that can be considered in membrane models.

**Figure 3 membranes-10-00310-f003:**
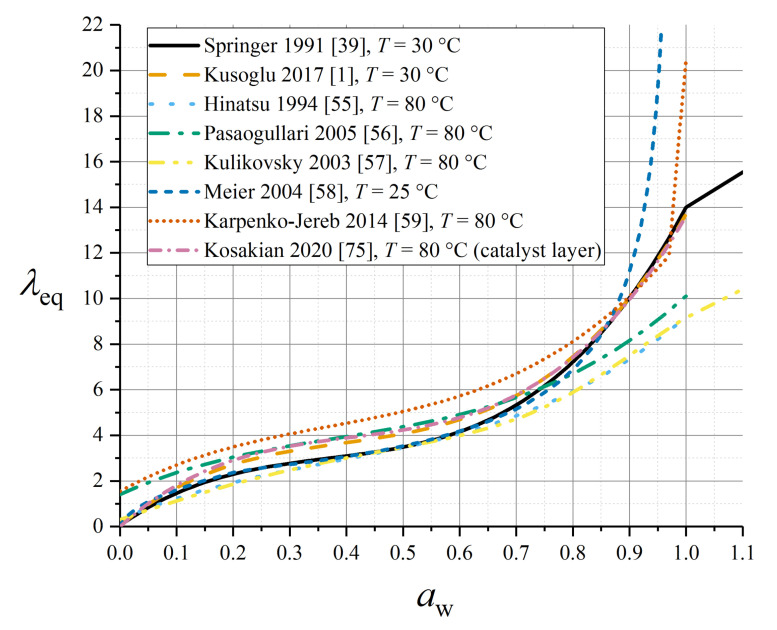
Summary of measured sorption isotherms for Nafion 1100 in the vapour-equilibrated range of *a*_w_.

**Figure 4 membranes-10-00310-f004:**
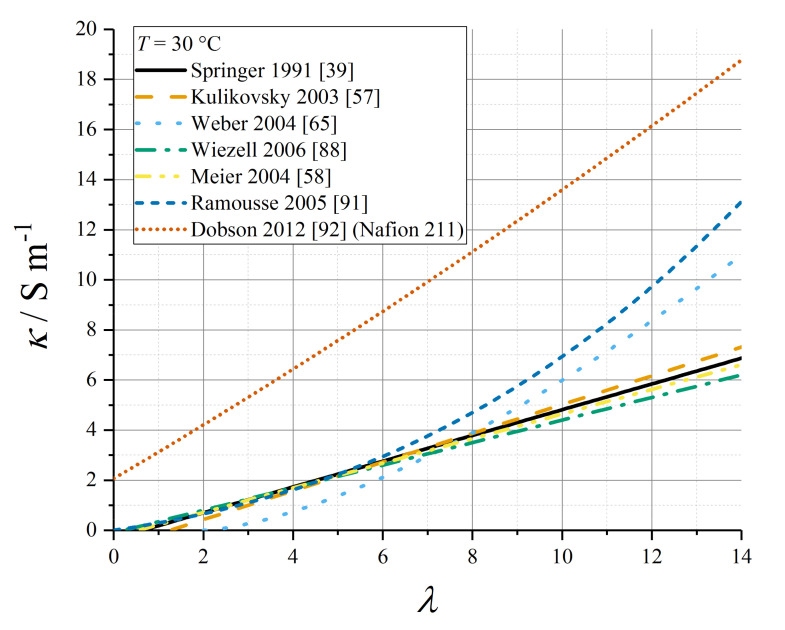
Summary of measured proton conductivity for Nafion 1100 in the vapour-equilibrated range of λ at room temperature (*T* = 30 °C).

**Figure 5 membranes-10-00310-f005:**
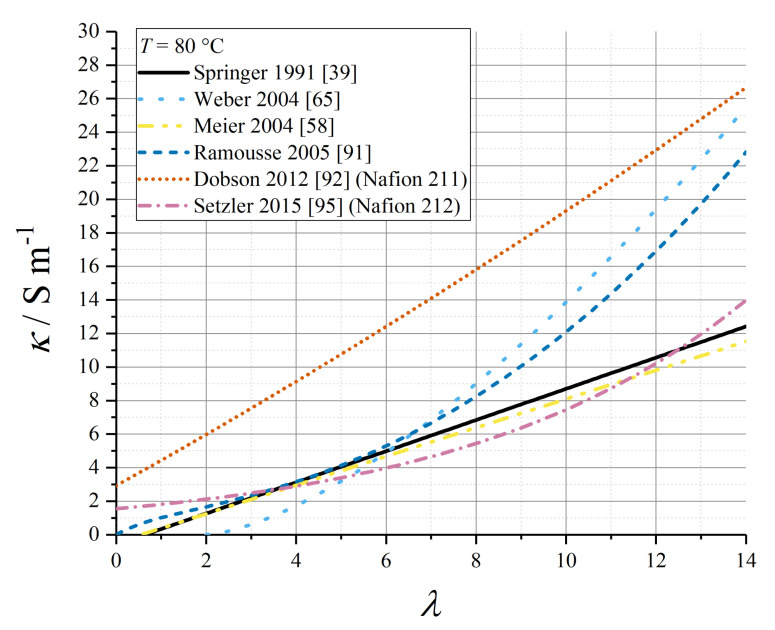
Summary of measured proton conductivity for Nafion 1100 in the vapour-equilibrated range of λ at typical PEMFC operating temperature (*T* = 80 °C).

**Figure 6 membranes-10-00310-f006:**
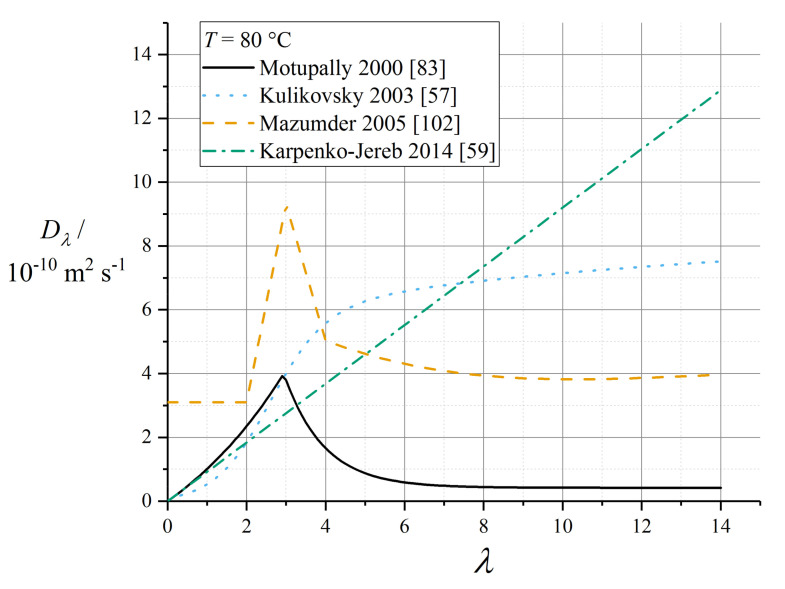
Summary of Fick’s law diffusivities of water in Nafion 1100, in the vapour-equilibrated range of *λ* at *T* = 80 °C.

**Figure 7 membranes-10-00310-f007:**
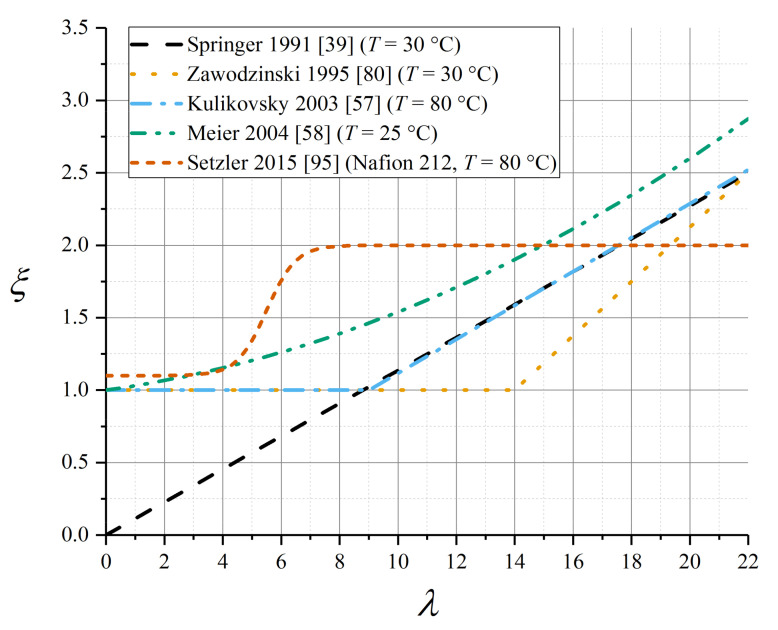
Summary of common parameterisations of electroosmotic drag coefficient for Nafion 1100.

**Figure 8 membranes-10-00310-f008:**
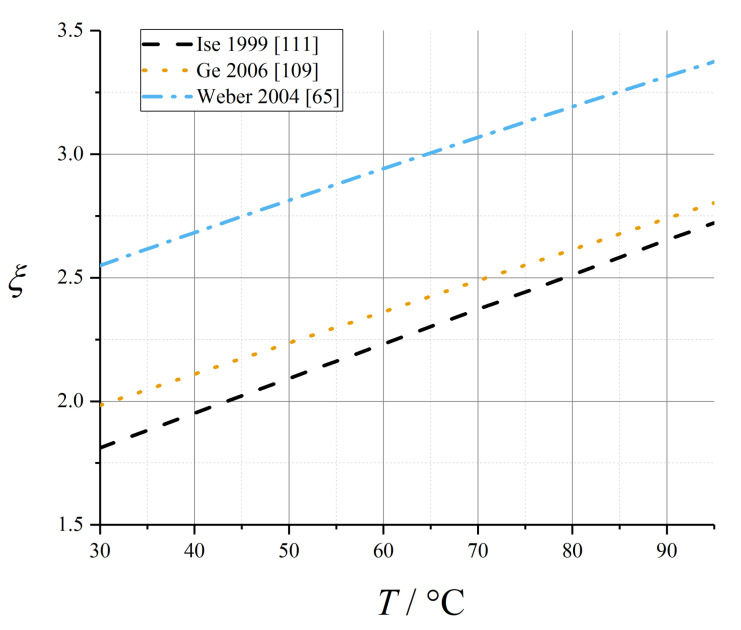
Temperature-dependent parameterisation of the liquid-equilibrated electroosmotic drag coefficient for Nafion 1100.

**Figure 9 membranes-10-00310-f009:**
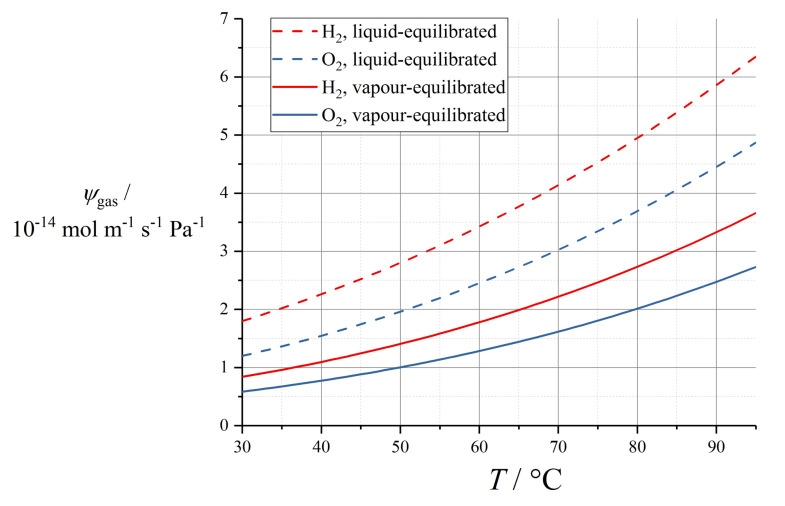
Permeation coefficients for H_2_ and O_2_ as a function of temperature, using data summarised by Weber and Newman [65].

**Figure 10 membranes-10-00310-f010:**
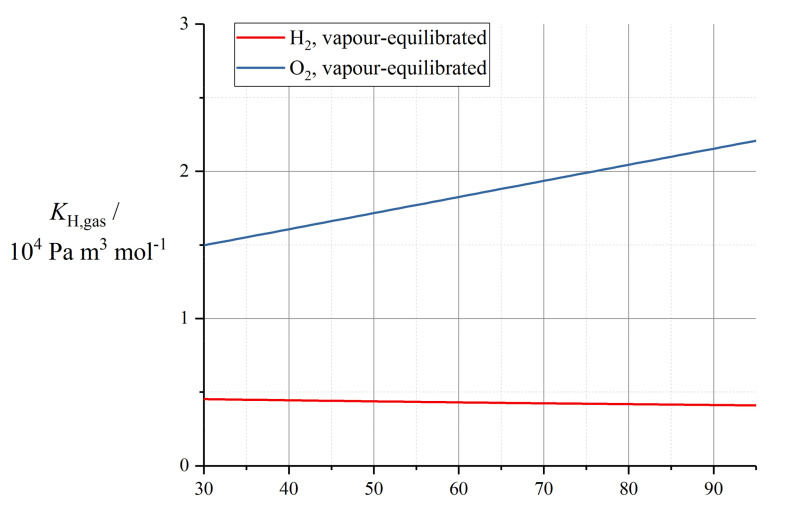
Henry’s law coefficients for H_2_ and O_2_ as a function of temperature, using data summarised by Wong and Kjeang [137].

**Table 1 membranes-10-00310-t001:** Parameterisation of the Springer et al. water vapour pressure fit, with *p*_0_ = 1 atm and *T*_0_ = 0 °C [39].

Coefficient	Value
*a* _0_	−2.1794
*a* _1_	+0.02953 K^−1^
*a* _2_	−9.1837 × 10^−5^ K^−2^
*a* _3_	+1.4454 × 10^−7^ K^−3^

**Table 2 membranes-10-00310-t002:** Parameterisation of the American Society of Heating, Refrigerating and Air-Conditioning Engineers (ASHRAE) water vapour pressure fit reported by Gurau et al. [49].

Coefficient	Value
*b* _−1_	−5.8002206 × 10^3^
*b* _0_	1.3914993
*b* _1_	−0.048640239
*b* _2_	4.1764768 × 10^−5^
*b* _3_	1.4452093 × 10^−8^
*b_e_*	6.5459673

**Table 3 membranes-10-00310-t003:** Parameterisation of the Meyers–Newman sorption model for Nafion 1100 at *T* = 30 °C [52,61].

Coefficient	Value
*K* _1_	100
*K* _2_	0.217
*E* _00_	−0.0417 kg mol^−1^
*E* _+0_	−0.052 kg mol^−1^
*E* _++_	−3.7216 kg mol^−1^

**Table 4 membranes-10-00310-t004:** Parameterisation of the binary friction model for Nafion 1100 [78,82].

Coefficient	Value
*λ* _min_	1.65
*D* _+0_	6.5 × 10^−9^ m^2^ s^−1^
*s*	0.83
*q*	1.5
*A* _+_	0.084
*A* _0_	0.5
*θ* _diff_	1800 K

**Table 5 membranes-10-00310-t005:** Summary of polynomial proton conductivity relations, with *T*_0_ = 303.15 K.

Data Source	*κ*_0_/S m^−1^	λ_0_	*θ*_cond_/K	*n* _cond_
Springer [39]	0.5139	0.6344	1268	1
van Bussel–Kulikovsky [57,87]	0.5736	1.253	undefined	1
Weber [65]	0.2646	2	1800	1.5
Wiezell [88], fit to Zawodzinski [52]	0.45	0.222	undefined	1
Meier [58], fit to Zawodzinski [79]	0.491 *	0.543	1190	1

* Original value is 0.46 at *T*_0_ = 298.15 K.

**Table 6 membranes-10-00310-t006:** Parameterisation of the catalyst layer proton conductivity model given by Kosakian et al., *T* = 80 °C [75].

*i*	*a_i_*	*b_i_*
0	−0.8	−0.1254
1	0.075	0.1832
2	−6.375 × 10^−4^	−8.65 × 10^−3^
3	1.93 × 10^−5^	9.4 × 10^−5^

**Table 7 membranes-10-00310-t007:** Parameterisation of linear expressions for the electroosmotic drag coefficient in the liquid-equilibrated regime.

Data Source	*λ* _crit_	*α_ξ,λ_*
Zawodzinski et al. [80] (room temperature)	14	0.1875
van Bussel–Kulikovsky (*T* = 80 °C) [57,87]	9	0.117

**Table 8 membranes-10-00310-t008:** Parameterisation of temperature dependence of the electroosmotic drag coefficient for Nafion 1100, with *T*_0_ = 303.15 K.

Data Source	*ξ* _0_	*α_ξ,T_*/K^−1^
Ge et al. [109]	1.984	0.0126
Ise et al. [111]	1.812	0.014

**Table 9 membranes-10-00310-t009:** Gas permeability data in Nafion 1100 as summarised by Weber and Newman, *T*_0_ = 303.15 K [65].

Coefficient	H_2_	O_2_
*ψ*_V0_/mol m^−1^ s^−1^ Pa^−1^	2.9 × 10^−15^	1.1 × 10^−15^
*ψ*_w_/mol m^−1^ s^−1^ Pa^−1^	2.2 × 10^−14^	1.9 × 10^−14^
*ψ*_L0_/mol m^−1^ s^−1^ Pa^−1^	1.8 × 10^−14^	1.2 × 10^−14^
*E*_A,*D*V_/kJ mol^−1^	21	22
*E*_A,*D*L_/kJ mol^−1^	18	20

**Table 10 membranes-10-00310-t010:** Henry’s law data for various gases in Nafion 1100 as summarised by Wong and Kjeang [137].

	*K*_H,gas,0_/Pa m^3^ mol^−1^	*θ*_soln,gas_/K
H_2_	2.584 × 10^3^	−170
O_2_	1.348 × 10^5^	666
HF	4.149 × 10^8^	7400
H_2_O_2_	6.83 × 10^7^	7379

**Table 11 membranes-10-00310-t011:** Gas diffusion coefficients in Nafion 1100 as reported by Bernardi and Verbrugge [142].

Gas	*D*_gas,0_/m^2^ s^−1^	*θ*_diff,gas_/K
H_2_	4.1× 10^−7^	2602
O_2_	3.1 × 10^−7^	2736

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
