# Peer review of "Modelling the Proton-Conductive Membrane in Practical Polymer Electrolyte Membrane Fuel Cell (PEMFC) Simulation: A Review"

_membranes, 2020, doi:10.3390/membranes10110310_

Round 1

Reviewer 1 Report

A review is presented in the article denominated "Modelling the Proton-Conductive Membrane in Practical Polymer Electrolyte Membrane Fuel Cell (PEMFC) Simulation", the review sounds interesting, but we have the following observations:

  1. In line 41, the authors said that the review will focus on the macroscopically observable transport behaviour of the membrane, but this is not indicated in the abstract.

  1. Many words alongside the manuscript, such as line 92, 180, 208, 228…, were written with another font. Can the authors explain the reason of these changes?

  1. I suggest to the authors to add some references for the Aquivion and 3M materials of the PFSA (line 103).

  1. Although the symbolic notations are defined at the end of the manuscript, they also should be defined when used. It is difficult to follow and understand the equations since the parameters and symbols are not defined within their equations.

  1. Please add the expression of the open circuit cell voltage EOCV (equation 3.1). How can the authors obtain the tafel slop Acat and the reference current density iref?

  1. Which references were used to define equation 4.1, 4.2, 4.3, 4.6, 4.7…? Please add the required references.

  1. What is P in equation 4.24?

  1. Can the authors explain the features of the sorption isotherm based on Springer et al. (since it is the most commonly one) over the one that defined by Kulikovsky?

  1. According to equation 4.33, the catalyst layer isotherm presented by Kosakian et al. gives a value of 22 for aw > 1, but this is not presented in figure 3.

  1. The empirical relations for the proton conductivity are presented in figure 4 (T=30°C) and figure 5 (T=80°C). Can the authors explain how can they obtain the curve of Kuliskovsky and Wiezell since θcond is not given in table 5. Besides, why these two curves are presented only for T=30°C?

  1. In line 643, what do the authors mean by m=L,V ?

  1. In line 682, D0,w should be D0,λ.

  1. In equation 5.30, 5.31, 5,32 and 5,34, the authors defined the effective diffusion coefficient Dλ but then, they plotted the Fick’s law diffusion coefficient Dw in figure 6. What is the relationship between Dλ and Dw?

  1. I suggest to the authors to plot the electroosmotic drag coefficient ξ=f(T) presented by Ge et al. and Ise et al. (fully liquid-equilibrated limit), as well as the electroosmotic drag coefficient of the liquid-equilibrated membrane which presented by Weber and Newman.

  1. I suggest to the authors to plot the permeation coefficient under vapour ψi,v=f(T) and liquid ψi,L=f(T) equilibrated conditions for H2 and O2. The same suggestion for The Henry’s law coefficient KH,gas=f(T) and gas diffusion coefficient Dgas=f(T).

Reviewer 2 Report

Summary:

This review article entitled “Modelling the Proton-Conductive Membrane in Practical Polymer Electrolyte Membrane Fuel Cell (PEMFC) Simulation: A Review” reports a comprehensive summary of the continuous development of in the theoretical models of transport phenomena of polymer electrolyte membranes, as applied to practical models of PEM fuel cells from 1991 till now. In total, 193 research articles on this research field are reviewed. The calculation models and the parameterization reported are compiled into a single source with consistent notation for showing the research trend. The formulation and parameterization of various models are summarized with the incorporation of different conditions and assumption. A prediction of the membrane degradation and why this affects the performance of the PEMFC is summarized in the last section.

General comment:

In general, this submission gives a comprehensive paper review showing the 30 years of development in the PEMFCs. A few comments below are listed for the authors as a reference during the revision. Thank you.

Comments:

1. General: Thanks for giving the Table A1. Table of Abbreviations. One minor suggestion is to review all abbreviations given and make sure they are consistently used. For example, “PEMFC” should be applied for “polymer electrolyte membrane fuel cell” and “PEM fuel cell”

2. General: The summary of all simulation work reported in these 30 years in PEMFC should give a perspective and future directions. The conclusion section should be reported as an individual section for showing the key comments. Thus, the current “Conclusion and perspective” is suggested to separately into a “perspective and future directions” and a “conclusion”

3. References: It is understandable that the references are all classic and old researches because of the theoretical discussion made here. However, it is better to make a survey in newly published work.

4. General: The data presented in Figures 3 – 7 seem to be collected from different reference papers. It is suggested to mark these citations with their reference number rather than the author name and publication year.

Thank you for considering my reviewer comments. Hope this will help.
